# DCAF15 control of cohesin dynamics sustains acute myeloid leukemia

Grant P. Grothusen ®[1,5], Renxu Chang[1,5], Zhendong Cao[1], Nan Zhou[1], Monika Mittal[1], Arindam Datta[1], Phillip Wulfridge[2], Thomas Beer[3], Baiyun Wang ®[4], Ning Zheng ®[4], Hsin-Yao Tang ®[3], Kavitha Sarma ®[2], Roger A. Greenberg ®[1], Junwei Shi ®[1] & Luca Busino ®[1]✉

The CRL4-DCAF15 E3 ubiquitin ligase complex is targeted by the aryl-sulfonamide molecular glues, leading to neo-substrate recruitment, ubiquitination, and proteasomal degradation. However, the physiological function of DCAF15 remains unknown. Using a domain-focused genetic screening approach, we reveal *DCAF15* as an acute myeloid leukemia (AML)-biased dependency. Loss of *DCAF15* results in suppression of AML through compromised replication fork integrity and consequent accumulation of DNA damage. Accordingly, *DCAF15* loss sensitizes AML to replication stress-inducing therapeutics. Mechanistically, we discover that DCAF15 directly interacts with the SMC1A protein of the cohesin complex and destabilizes the cohesin regulatory factors PDS5A and CDCA5. Loss of PDS5A and CDCA5 removal precludes cohesin acetylation on chromatin, resulting in uncontrolled chromatin loop extrusion, defective DNA replication, and apoptosis. Collectively, our findings uncover an endogenous, cell autonomous function of DCAF15 in sustaining AML proliferation through post-translational control of cohesin dynamics.

The ubiquitin-proteasome system (UPS) is fundamental in the control of cellular function, and its dysregulation has been implicated in various diseases, including cancers[1,2]. The clinical success of the proteasome inhibitor bortezomib/Velcade[3–5] and E3 ubiquitin ligase molecular glue activator IMiDs[6–8] and aryl-sulfonamides[9–11] in the treatment of hematological malignancies has highlighted the importance of understanding the biological significance of protein ubiquitination and degradation mechanisms. However, despite the large number of E3 ligases (>600), only a limited number have well-characterized substrates and biological functions[2].

One of the largest families of E3 ligases are the Cullin-RING Ligases (CRLs). CRLs consist of a Cullin scaffold protein, an adaptor protein for docking substrate receptor subunits to the Cullin, and a RING-domain-containing protein for recruiting the E2[12,13]. DCAF15 (DDB1- and Cul4-Associated Factor 15) is a substrate receptor subunit of the CRL4

complex. DCAF15 has garnered substantial therapeutic interest in cancer, as it has been shown to interact with the anti-cancer aryl-sulfonamide molecular glue compounds (including the drugs indisulam [also known as E7070], tasisulam, E7820, and chloroquinoxaline sulfonamide [CQS]), which enables the E3 to recognize and promote ubiquitin-dependent degradation of the splicing factor RNA-Binding Motif protein 39 (RBM39)[9,10,14–16]. Accordingly, indisulam-mediated RBM39 depletion by DCAF15 results in altered splicing of HOXA9 target genes, and cytotoxicity in cancers including acute myeloid leukemia (AML)[17]. Little however is known about the physiological molecular function of DCAF15, and a bona fide endogenous target of DCAF15 has remained elusive[18].

Cohesin is a multi-subunit ring-shaped protein complex that plays vital roles in diverse cellular processes, including sister chromatid cohesion, chromatin looping and genome architecture, DNA damage

[1]Department of Cancer Biology, Perelman School of Medicine, University of Pennsylvania, Philadelphia, PA, USA. [2]Ellen and Ronald Caplan Cancer Center, The Wistar Institute, Philadelphia, PA, USA. [3]Proteomics and Metabolomics Facility, The Wistar Institute, Philadelphia, PA, USA. [4]Howard Hughes Medical Institute, Department of Pharmacology, University of Washington, Seattle, WA, USA. [5]These authors contributed equally: Grant P. Grothusen, Renxu Chang. ✉e-mail: businol@upenn.edu

response, and DNA replication fork progression[19,20]. It is a frequently mutated protein complex in cancer[19], and in particular, recurrent mutations in several subunits have been identified in myelodysplastic syndrome (MDS) and AML[21,22]. Importantly, mutations in cohesin proteins have been associated with aberrant DNA damage repair, increased genomic instability, and aberrant DNA loop extrusion[23]. Structurally, the cohesin ring consists of four core subunits: SMC1A, SMC3, RAD21, and STAG1 or STAG2. The head regions of the SMC1A and SMC3 proteins contain an ATP-binding cassette (ABC) family ATPase domain which binds and hydrolyzes ATP, promoting head association/dissociation and DNA loading/unloading[24–27]. Extensive evidence suggests that cohesin loading is coupled with DNA replication[28]. Key regulatory roles are played by the acetyltransferases ESCO1 and ESCO2 and the accessory cohesin-binding proteins PDS5A and PDS5B, CDCA5 (Sororin), and WAPL. ESCO1/ESCO2 acetylates SMC3 on two conserved lysines, promoting association of PDS5A/CDCA5 with the cohesin complex to stabilize cohesin-DNA interaction[29]. CDCA5 maintains cohesion by displacing WAPL, a cohesin removal factor, from PDS5A[29–31]. The cycle of cohesin loading and unloading from chromatin is critical for DNA replication[32], as well as for 3D genome organization[33,34], however comprehensive identification of the factors regulating cohesin complex recycling has not been achieved.

Here, we identify DCAF15 as a critical factor for cell proliferation in AML and unravel its role in controlling cohesin acetylation and dynamics on DNA via promoting PDS5A/CDCA5 protein clearance at the cohesin complex.

## Results

### DCAF15 is an AML-biased E3 ubiquitin ligase dependency

To identify genes in the UPS pathway that sustain cancer proliferation, we performed domain-focused CRISPR/Cas9-based knockout (KO) negative selection (dropout) screens in the following human cancer cell lines: MOLM-13 (AML), MV4-11 (AML), Jurkat (T-cell Acute Lymphoblastic Leukemia [T-ALL]), U-2932 (Diffuse Large B-cell Lymphoma [DLBCL])[35], OPM-1 (Multiple Myeloma [MM]), and Hep-G2 (Hepatocellular Carcinoma [HCC]). Here, we utilized an in-house curated single-guide RNA (sgRNA) library targeting the domain in the Cullin-RING Ligase (CRL) receptor that connects the receptor to the Cullin module (Fig. 1a and Supplementary Data 1). An initial timepoint of pooled, edited cells was collected 3 days after library transduction, for comparison of sgRNA barcode abundance with cells collected after 5 population doublings at the experimental endpoint, and an average CRL domain Essentiality Score (ES)[36] was calculated for each CRL gene in each cell line (Fig. 1b).

To identify AML-biased CRL dependencies, the screened CRL genes were ranked according to the difference in ES values between AML cell lines and non-AML cell lines (Fig. 1c, Supplementary Fig. 1a–c, and Supplementary Data 1), and FBXO11 and DCAF15 were revealed as top AML-dependency genes. While mRNA expression of the top hit, FBXO11, did not correlate with overall survival (OS) in the TCGA-LAML patient dataset (Supplementary Fig. 1d), high expression of DCAF15 correlated significantly with poorer OS (Fig. 1d). Furthermore, DCAF15 mRNA expression was significantly higher in AML patient samples compared to normal hematopoietic stem cells (HSC) (Fig. 1e), and significantly higher in AML cell lines compared to solid tumor cell lines according to the Cancer Cell Line Encyclopedia (CCLE)[37] (Supplementary Fig. 1e), indicating a potential cancer-promoting function of DCAF15 in AML.

To validate the requirement of DCAF15 in AML cellular proliferation, a competition-based proliferation assay was performed. MV4-11 cells transduced with 3 different sgRNAs targeting DCAF15 were outcompeted by parental cells and depleted, validating our pooled sgRNA library screening results (Fig. 1f). Since no effective commercial antibody against the DCAF15 protein exists, depletion of DCAF15 protein

was confirmed by loss of indisulam-dependent degradation of RBM39[9,10] (Supplementary Fig. 1f). Validating the on-target effect of DCAF15-targeting sgRNAs, a CRISPR-resistant DCAF15 cDNA ectopically expressed (Supplementary Fig. 1g) in MV4-11 DCAF15−/− cells was capable of partially rescuing indisulam-mediated RBM39 degradation (Fig. 1g) and the proliferation defect upon DCAF15 loss (Fig. 1h), likely attributable to restoration of partial ligase activity of DCAF15. Additionally, non-competitive proliferation rate assessments conducted in pure sorted populations of control and DCAF15-knockout MV4-11 and OCI-AML3 AML cells confirmed an overall slowing of cell division in AML cells lacking DCAF15 (Supplementary Fig. 1h).

Taken together, these results revealed a pro-proliferative function of DCAF15 in AML.

### DCAF15 loss suppresses AML via activation of p53

To gain insights into the cell autonomous function of DCAF15, we investigated the effect of DCAF15 ablation on the AML transcriptome via RNA-seq. MV4-11 cells were transduced with an sgRNA targeting DCAF15 and sorted/collected 4 days after infection. Differential expression analysis of the mRNA reads revealed that 18 and 24 genes were significantly down-regulated or up-regulated in DCAF15-knockout cells, respectively (Fig. 2a and Supplementary Data 2). Interestingly, analysis of enriched gene sets using the Enrichr[38] platform revealed the p53 pathway as the most significantly up-regulated transcriptional signature upon DCAF15-knockout (Fig. 2b and Supplementary Data 2). Furthermore, analysis of predicted regulatory features and cis-regulatory modules also revealed enrichment in binding sites for the p53 transcription factor as the top hit (Fig. 2c and Supplementary Data 2). p53 activation in DCAF15-knockout cells was confirmed by increased levels of p21 (CDKN1A, a well-characterized p53 target gene[39]) and cleaved caspase-3 (indicative of apoptosis activation) (Fig. 2d). In an orthogonal approach, OCI-AML3 cells were edited by inserting a FLAG-HA-tagged FKBP12(F36V) sequence (dTAG cassette[40]) at the start codon of the endogenous DCAF15 locus (dTAG-DCAF15) and assessed for degradation of DCAF15 upon dTAG-V1 treatment (Supplementary Fig. 2a). Using this model, we conducted RNA-sequencing analysis upon dTAG-V1 treatment for 24 h and 72 h (Supplementary Fig. 2b and Supplementary Data 2). Similar to what was observed in DCAF15-knockout cells, acute DCAF15 degradation led to upregulation of the p53 pathway (Supplementary Fig. 2c).

These findings suggested that the transcriptional changes induced by DCAF15 loss could be rescued by concomitant TP53-ablation. Indeed, RNA-seq demonstrated that only 5 and 1 protein-coding mRNAs were significantly down-regulated or up-regulated, respectively, in TP53/DCAF15 double-knockout (DKO) AML cells (Fig. 2e, Supplementary Fig. 2d, e, and Supplementary Data 2). Importantly, individual TP53 target genes, including CDKN1A, were up-regulated at the mRNA level upon DCAF15 loss in a p53-dependent manner (Supplementary Fig. 2f).

Next, we tested whether the TP53-mediated transcriptional changes in DCAF15-knockout cells could account for the pro-proliferative effect of DCAF15. AML cells transduced with two different sgRNAs targeting TP53 largely rescued the proliferation defect induced by DCAF15 loss that was mediated by transduction with two different sgRNAs targeting DCAF15 (Fig. 2f). Lack of complete proliferation rescue may suggest that DCAF15 possesses p53-independent pro-proliferative functions as well. In line with the observation that protein levels of cleaved caspase-3 were elevated in DCAF15-knockout cells (Fig. 2d), apoptosis, measured by Annexin-V staining, was significantly up-regulated upon DCAF15 loss in TP53-WT AML cells; however, concomitant TP53-ablation rescued the observed apoptotic phenotype (Fig. 2g).

These data led us to question whether DCAF15 may be a dependency specifically in TP53-WT cases of AML. To test this hypothesis, we conducted proliferation competition assays in a panel

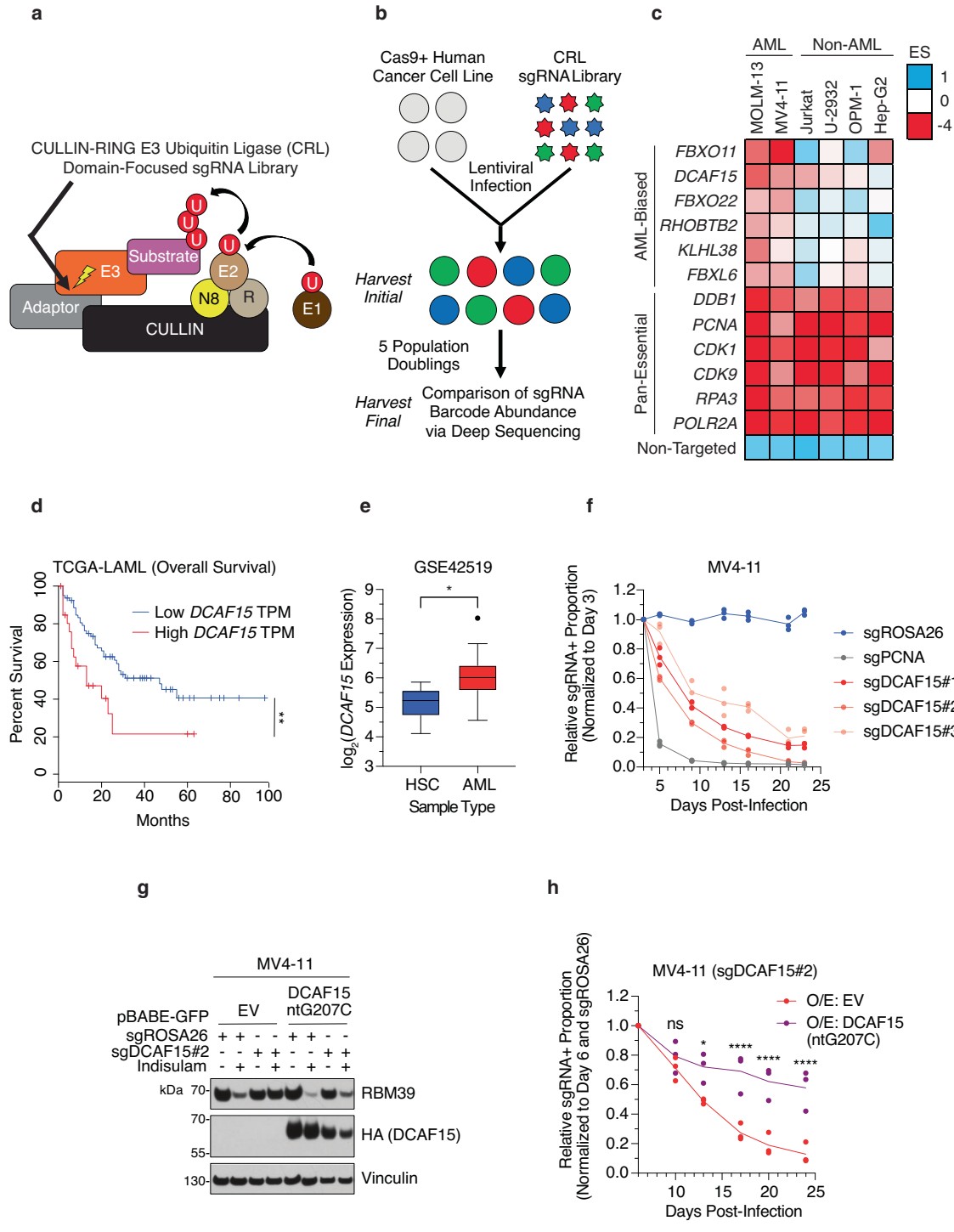

of 7 human AML cell lines: 4 in which p53 activity is abrogated by *TP53* inactivating mutations (THP-1, SET-2, HEL, U-937), and 3 in which *TP53* is wild-type (MOLM-13, OCI-AML3, MV4-11). Strikingly, DCAF15 loss resulted in the depletion of all *TP53*-WT cell lines at a higher rate than all *TP53*-mutant cell lines; this effect was specific to DCAF15 as the Cas9 editing efficiency was similar in all cell lines tested (Fig. 2h, Supplementary Fig. 2g, h). Accordingly, apoptosis was significantly up-regulated in *TP53*-WT AML cells, but not in *TP53*-mutant cells (Fig. 2i).

Altogether, these results demonstrated that DCAF15 promotes AML via suppression of p53 and that the growth defect of AML cells lacking DCAF15 may be attributable to a combination of both reduced cell proliferation rate as well as induction of apoptosis.

## DCAF15 interacts with the cohesin complex

Given its function as an E3 ubiquitin ligase, we hypothesized that DCAF15 sustains AML proliferation through promoting the degradation of specific target substrates. In order to rule out the possibility that DCAF15 degrades p53 directly, we constructed *TP53*-WT AML cells expressing a doxycycline (Dox)-inducible shRNA targeting *DCAF15*. Analysis of the p53 protein half-life revealed no difference between cells expressing or lacking DCAF15, indicating that DCAF15 is not involved in p53 degradation (Supplementary Fig. 3a).

Next, we utilized a dual approach based on unbiased identification of protein association by comparing the DCAF15-immunoprecipitation-interacting proteome to the DCAF15-proximity-labeling proteome (Supplementary Fig. 3b). First, mass spectrometry

**Fig. 1 | DCAF15 is an AML-biased E3 ubiquitin ligase dependency. a** Schematic describing the design of CULLIN-RING E3 ubiquitin ligase (CRL) domain-focused sgRNA library. If known, sgRNAs targeted the receptor-adaptor interacting domain of the E3 (indicated with arrow), otherwise targeted first exon of gene. N8 NEDD8, R RING, U Ubiquitin. **b** Schematic describing workflow for CRISPR dropout screen using CRL sgRNA library. Protein domain Essentiality Score (ES) calculated for each gene as average[$\log_2$(final sgRNA abundance +1/initial sgRNA abundance)]. **c** Heatmap results for selected CRL genes and controls. Genes ranked by AML-biased ES, defined by the difference of ES in AML versus non-AML cell lines. **d** Overall survival of AML patients using GEPIA platform[83] with low (bottom 75%) or high (top 25%) DCAF15 mRNA expression (TCGA-LAML dataset [https://www.cancer.gov/tcga]). TPM Transcripts Per Million. $n = 106$. log-rank test, **$p$-value < 0.01 ($p = 0.0069$). **e** DCAF15 mRNA expression in $n = 252$ AML patient samples compared to $n = 6$ normal hematopoietic stem cells (HSC) samples according to microarray expression profiling from GSE42519[84,85], probe 91952_at. Welch's two-sided $t$-test, *$p$-value < 0.05 ($p = 0.0193$). Center line shows the median, box limits show 75th and 25th percentiles, whiskers show minimum-maximum values, with the outlier shown as independent dot (Tukey method). Generated using BloodSpot platform[86]. **f** Competition-based proliferation assay performed in Cas9 + MV4-11 cell line. sgRNA+ populations monitored over time for mCherry expression by flow cytometry. sgRNA+ proportions normalized to day 3 post lentiviral sgRNA infection monitored over 23 days. $n = 3$ biologically independent replicates. sgROSA26, negative control; sgPCNA, positive control. **g** MV4-11 Cas9+ cell lines stably expressing empty vector (EV) or HA-tagged CRISPR-resistant DCAF15[ntG207C] were infected with ROSA26-targeting (negative control) or DCAF15-targeting sgRNAs. Cells treated with DMSO or 3 μM indisulam for 6 h and lysates analyzed by Western blot for indicated proteins. Immunoblots representative of two independent experiments. **h** Competition-based proliferation assay performed in MV4-11 Cas9+ cell lines stably expressing empty vector (EV) or DCAF15[ntG207C] and infected with ROSA26-targeting (negative control) or DCAF15-targeting sgRNAs. sgRNA+ populations monitored over time for mCherry expression by flow cytometry. sgRNA+ proportions normalized to day 6 post lentiviral sgRNA infection monitored over 24 days. $n = 3$ biologically independent replicates. Two-way ANOVA (with Bonferroni's multiple comparisons test), ****padj < 0.0001, *padj < 0.05 (padj = 0.021), ns not-significant. Source data are provided as a Source Data file.

analysis of FLAG-immunoprecipitants from transiently over-expressing FLAG-tagged DCAF15 HEK293T cells revealed 395 proteins with unique peptides enriched at least 5-fold in cells over-expressing DCAF15 compared to an empty vector (EV) (Fig. 3a and Supplementary Data 3). In the second approach, we generated MV4-11 cells stably expressing TurboID-DCAF15. To Identify bona fide substrates, cells were treated with or without MLN4924, a NEDD8-activating enzyme (NAE) inhibitor which blocks Cullin protein neddylation and CRL-mediated ubiquitination[41,42], resulting in accumulation of biotinylated interactors of DCAF15. Streptavidin pull-down of cell lysates pre-treated with biotin was analyzed by mass spectrometry and peptides ratio (MLN4924/DMSO) revealed 60 proteins enriched at least 5-fold in MLN4924-treated cells (Fig. 3b and Supplementary Data 3). To narrow the list of substrate candidates, we overlapped the immunoprecipitated FLAG-DCAF15-interactome with the proximity-labeling dataset; 4 proteins were identified: VprBP, SMC3, SMC1A, and MCM4 (Fig. 3c). Interestingly, 2 of these proteins (SMC3 and SMC1A) are core components of the cohesin complex[43].

Further analysis of the immunopurified and proximity-labeling datasets revealed that DCAF15 interacted with the core cohesin ring subunits SMC1A and SMC3 (FLAG-proteome, Fig. 3d), however it also positioned in proximity to RAD21 and STAG2, as well as accessory cohesin-binding proteins PDS5A, PDS5B, WAPL, and CDCA5 (TurboID-proteome, Fig. 3d). These unbiased data were validated in downstream immunoprecipitation assays in cells (Fig. 3e) and in vitro (Supplementary Fig. 3c), as well as in proximity-labeling affinity purification (Fig. 3f).

Homology sequence analysis revealed a conserved internal loop region of DCAF15 (amino acids 274-382), which we hypothesized to be important in the recognition of an endogenous target. Indeed, FLAG-immunoprecipitation of DCAF15($\Delta$274-382) mutant resulted in loss of interaction with SMC1A and SMC3; accordingly, immunoprecipitation of the internal loop region only, DCAF15(aa274-382), was sufficient to promote interaction with SMC1A and SMC3 (Supplementary Fig. 3d). Interestingly, the DCAF15 internal loop region differs from the region required for DCAF15 molecular glue-mediated neo-substrate binding[16], implying that the cohesin complex may not be affected by aryl-sulfonamide molecular glue compounds.

SMC1A and SMC3 proteins are composed of three structural domains: (1) a hinge domain mediating the SMC1A-SMC3 interaction, (2) a head domain mediating the SMC-RAD21 interaction, and (3) a coiled-coil domain connecting the SMC hinge and head domains[43]. Immunoprecipitation of individual FLAG-SMC1A domains in HEK293T cells stably expressing HA-DCAF15 demonstrated that the SMC1A head (RAD21-binding) domain is both necessary and sufficient for binding to DCAF15 (Fig. 3g). AlphaFold[44] prediction of the DCAF15-

SMC1A head interaction revealed a high confidence structure in which a DCAF15 continuous 23 amino acids α-helix (aa315-337) interacts with SMC1A through an extensive hydrophobic interface substantiated by polar interactions (Fig. 3h). The DCAF15 α-helix makes direct contacts to both the N-terminal and C-terminal domains of the SMC1A head, and the DCAF15-SMC1A interface relies on the close interaction between the two halves. AlphaFold prediction of a potential DCAF15-SMC3 interface failed to yield a solution with a high confidence score, suggesting that while DCAF15 is brought into proximity with the entire cohesin complex, it physically interacts with SMC1A.

Superposition of the predicted DCAF15-SMC1A structure and the published cohesin complex structure 6WG3[45] revealed that DCAF15 and the C-terminal domain of RAD21 bind at the same site on SMC1A (Supplementary Fig. 3e). Consistently, truncation of the C-terminal domain of SMC1A resulted in loss of both DCAF15 and RAD21 inter-actions to SMC1A (Supplementary Fig. 3f). Interestingly, the DCAF15 α-helix mimics the action of a central α-helix of RAD21, which forms the predominant interface with SMC1A. Therefore, DCAF15 resembles RAD21 in its mode of interaction to the head domain of SMC1A. Based on the predicated interaction interface, we determined tyrosine 1204 (Y1204) in SMC1A to be important for the recruitment of DCAF15; indeed, its mutation weakened interaction with DCAF15 but not with RAD21 (Supplementary Fig. 3g).

Altogether, these data indicated that DCAF15 is brought into proximity with the cohesin complex via interaction with SMC1A.

## DCAF15 destabilizes cohesin-bound PDS5A and CDCA5

Despite the interaction, DCAF15 loss did not change the abundance of SMC1A, neither at the protein level (Supplementary Fig. 4a) nor at the mRNA level (Supplementary Fig. 4b). Hence, we hypothesized that DCAF15 could target for degradation other members of the cohesin complex when assembled into the ring-type structure. To test this hypothesis, we first questioned whether the formation of an intact cohesin complex is required for the recruitment of DCAF15. To this purpose, core cohesin ring subunits SMC3 (Fig. 4a) or RAD21 (Supplementary Fig. 4c) were knocked down with siRNAs prior to immunoprecipitation of FLAG-SMC1A and FLAG-CDCA5 in HEK293T cells stably expressing HA-DCAF15. Indeed, depletion of SMC3 or RAD21 decreased interaction of SMC1A and CDCA5 with DCAF15 (Fig. 4a, Supplementary Fig. 4c). Conversely, overexpression of RAD21 increased interaction of DCAF15 with SMC1A (Supplementary Fig. 4d). Furthermore, a mutation of CDCA5 in its FGF motif, which is required for interaction with PDS5A[29], impaired DCAF15 interaction (Supplementary Fig. 4e). Altogether, these data agree with the hypothesis that DCAF15 accesses cohesin proteins assembled on the intact ring-shape structure.

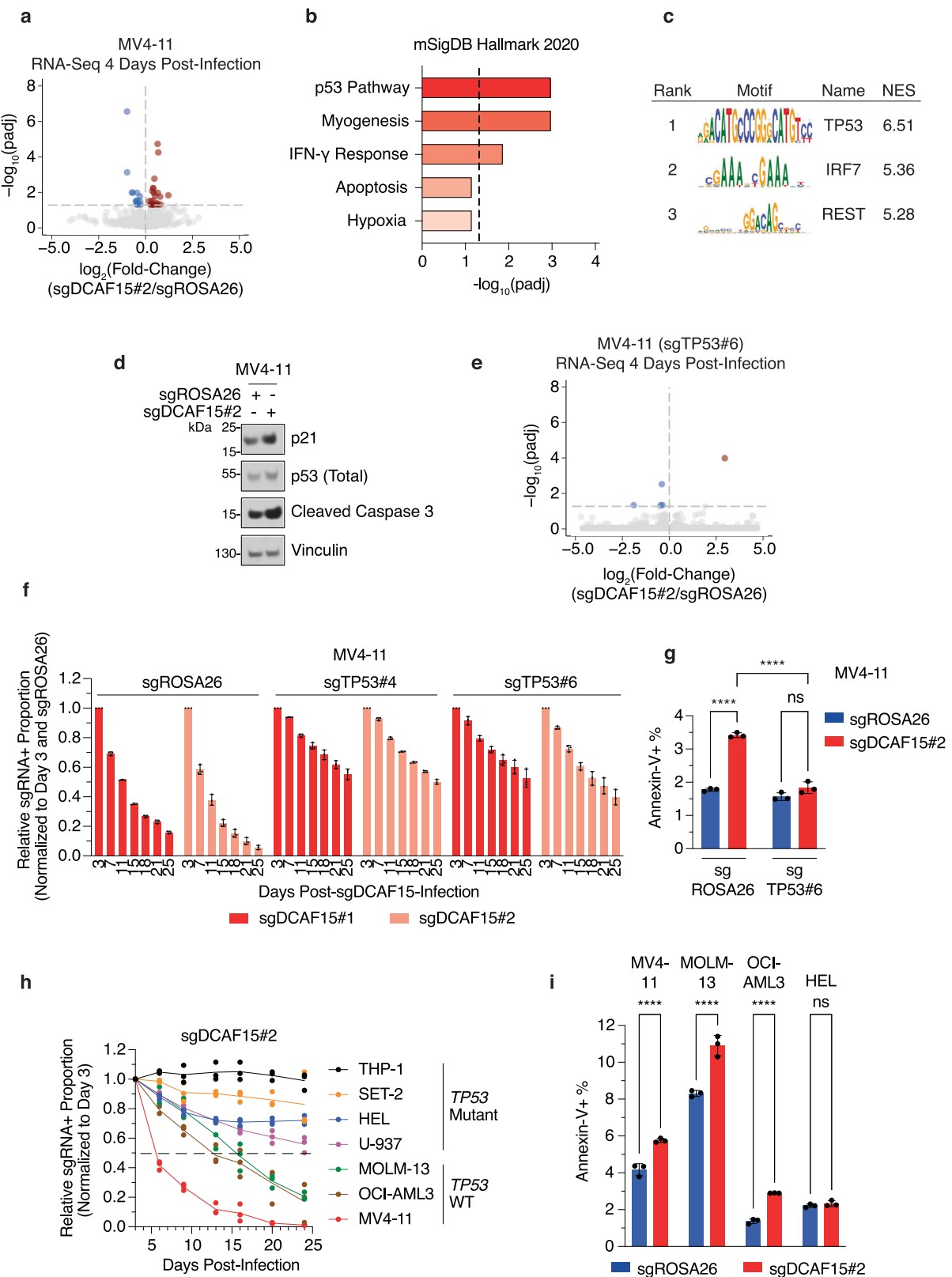

Next, we assessed whether the protein levels of intact cohesin ring complex members were regulated by DCAF15. As such, we purified the endogenous cohesin complex via immunoprecipitation of SMC1A in control versus *DCAF15*-knockout cells (Fig. 4b). Since *DCAF15* loss results in p53-dependent cell death, we utilized HEL cells, a *TP53*-mutant AML cell line, proliferation of which was not majorly affected by DCAF15 loss (Fig. 2h). We observed striking accumulation of PDS5A

and CDCA5 protein levels in the SMC1A-bound samples in cells lacking *DCAF15*, while binding of other core cohesin ring and accessory cohesin-binding protein subunits (SMC3, RAD21, PDS5B, WAPL) remained unchanged (Fig. 4b). Cycloheximide-chase assessment of the protein levels of PDS5A and CDCA5 bound to SMC1A also revealed extended half-lives of PDS5A, and to a lesser extent CDCA5, in *DCAF15*-knockout cells (Fig. 4c). Furthermore, inhibition of the proteasome or

**Fig. 2 | *DCAF15* loss suppresses AML via activation of p53. a** Differentially-expressed genes (sgDCAF15#2/sgROSA26) for MV4-11 (*TP53*-WT) cells revealed by RNA-seq. Shown in blue and red, respectively, are mRNAs significantly decreased or increased. *n* = 3 biologically independent replicates; DESeq2: two-sided Wald test adjusted with Benjamini and Hochberg method for multiple comparisons, padj < 0.05. **b** Enriched terms from MSigDB_Hallmark_2020 pathways gene set library[87] using Enrichr[38]. Top 5 enriched terms shown for genes up-regulated in MV4-11 sgDCAF15#2/sgROSA26 with padj < 0.05. $\log_{10}$(padj) scale. Dashed line indicates the cut-off for significantly enriched terms (padj < 0.05). **c** Transcription factors with enriched binding site motifs within genes significantly enriched (padj < 0.05) upon *DCAF15*-knockout in MV4-11 cells. Ranked using normalized enrichment score (NES). Generated using i-cisTarget platform[88]. **d** Lysates from Cas9 + MV4-11 cells infected with *ROSA26*-targeting or *DCAF15*-targeting sgRNAs, harvested 4 days post-infection, analyzed by Western blot for indicated proteins. Immunoblots representative of three independent experiments. **e** Same as in (**a**), except MV4-11 *TP53*-knockout (sgTP53#6) cells utilized. *n* = 3 biologically independent replicates; DESeq2: two-sided Wald test adjusted with Benjamini and Hochberg method for multiple comparisons, padj < 0.05. **f** Competition-based proliferation assays in Cas9 + MV4-11 *TP53*-WT (sgROSA26) and *TP53*-knockout (sgTP53#4 and sgTP53#6) cell lines. Plotted are relative sgRNA+ proportions normalized to day 3 sgRNA+ proportions and respective mCherry-co-expressed *ROSA26*-targeting proportions over 25 days. Error bars mean ± SD, *n* = 3 biologically independent replicates. **g** Cas9 + MV4-11 *TP53*-WT (sgROSA26) and *TP53*-knockout (sgTP53#6) cells infected with *ROSA26*- or *DCAF15*-targeting sgRNAs. On day 3 post-infection, cells stained for Annexin-V for quantification of early-apoptosis induction. Error bars mean ± SD, *n* = 3 biologically independent replicates. Two-way ANOVA (with Tukey's multiple comparisons test), ****padj < 0.0001; ns not-significant. **h** Competition-based proliferation assay performed in 7 different Cas9+ AML cell lines. Plotted are relative sgRNA+ proportions normalized to day 3 sgRNA+ proportions over 24 days. *n* = 3 biologically independent replicates. Dashed line indicates 50% proliferation defect passed during course of experiment. *TP53*-WT or *TP53*-Mutant status is indicated. **i** Cas9+ AML cells infected with *ROSA26*- or *DCAF15*-targeting sgRNAs. On day 3 post-infection, cells stained for Annexin-V for quantification of early-apoptosis induction. Error bars mean ± SD, *n* = 3 biologically independent replicates. Two-way ANOVA (with Tukey's multiple comparisons test), ****padj < 0.0001, ns not-significant. Source data are provided as a Source Data file.

CRL-mediated ubiquitination led to stabilization of PDS5A and CDCA5 bound to the cohesin complex in *DCAF15*$^{+/+}$ cells (Fig. 4d); however, in *DCAF15*-knockout cells, cohesin-bound PDS5A- and CDCA5-levels were already elevated and not further affected by proteasome or CRL inhibition (Fig. 4d). Interestingly, treatment with indisulam had no effect on the levels of PDS5A or CDCA5 (Fig. 4d), consistent with our binding data indicating that the mechanism by which DCAF15 controls its endogenous substrates differs from its molecular glue-mediated function (Supplementary Fig. 3d). Finally, an in vitro ubiquitination assay demonstrated that cohesin-bound DCAF15 is capable of catalyzing poly-ubiquitination of PDS5A, but not CDCA5, WAPL, or SMC1A (Fig. 4e), revealing PDS5A as the primary target of DCAF15-mediated ubiquitination. This finding also suggests that the increased stability of cohesin-bound CDCA5 in *DCAF15*-knockout cells could be a downstream result of the loss of DCAF15-dependent PDS5A ubiquitination and removal from the cohesin complex, supporting a model by which sustained interaction of CDCA5 with cohesin-bound PDS5A may indirectly shield it from APC-Cdh1-mediated degradation[46].

Altogether, these data suggested that DCAF15 destabilizes PDS5A and CDCA5 bound to the intact cohesin complex, via promoting ubiquitination and proteasomal degradation of PDS5A (Fig. 4f).

## DCAF15 sustains cohesin acetylation on chromatin

Acetylation of SMC3 at the Lysine-105 and Lysine-106 residues by the acetyltransferases ESCO1 and ESCO2 is critical for promoting the chromatin-bound state of cohesin[30,47] and the subsequent recruitment of CDCA5 to PDS5A[29,48,49]. As such, cohesin-bound to CDCA5/PDS5A maintains sister chromatid cohesion from S phase until mitosis[50]. Importantly, HDAC8-mediated SMC3-deacetylation is required for proper recycling of cohesin by initiating clearance of cohesin-bound accessory proteins (i.e., PDS5A/B and CDCA5)[32]. Interestingly, analysis of the Broad Institute Cancer Dependency Map (DepMap) revealed that *DCAF15* and the histone deacetylase *HDAC8* are top cancer co-dependencies (Supplementary Fig. 5a), suggesting that these proteins may function together in the same cohesin recycling process.

Taking advantage of the HDAC8-specific inhibitor PCI-34051[51], we found that inhibition of HDAC8 did not cooperate with DCAF15 loss in reducing cell viability, suggesting both genes function in the same pathway (Supplementary Fig. 5b). Analysis of FLAG-tagged DCAF15 immunoprecipitants from cells pre-treated with PCI-34051 revealed that DCAF15 does not interact with cohesin containing acetylated-SMC3 (Fig. 5a), suggesting that HDAC8-mediated SMC3-deacetylation precedes DCAF15 binding to the cohesin complex. Next, we questioned whether the absence of DCAF15-mediated PDS5A/CDCA5 removal might impact the ability of cohesin to get re-acetylated upon re-loading onto chromatin. We purified the cohesin complex via endogenous immunoprecipitation without or with PCI-34051 treatment. In *DCAF15*$^{+/+}$ cells, inhibition of HDAC8 led to SMC3-acetylation and increased association of CDCA5 and PDS5A with endogenous SMC1A (Fig. 5b), as recently published[52]. In *DCAF15*$^{-/-}$ cells, cohesin was hyper-loaded with CDCA5/PDS5A, and this association was not changed by HDAC8 inhibition (Fig. 5b). Additionally, compared to control cells, where inhibition of HDAC8 led to accumulation of acetylated-SMC3, *DCAF15*-knockout cells displayed a defect in SMC3-acetylation (Fig. 5b).

To assess the impact of *DCAF15* loss within distinct cell cycle phases, we utilized a cell line with endogenous tagging of *DCAF15* with *FLAG-HA-FKBP12(F36V)*. Cells were synchronized in G1/S, G2, or prometaphase, and the endogenous cohesin complex was immunopurified via SMC1A upon 6 h of dTAG-13 treatment. Cell cycle specific DCAF15-depletion resulted in the accumulation of cohesin-bound CDCA5/PDS5A at all cell cycle phases assessed, most strikingly during G2 and prometaphase (Fig. 5c); no defect in SMC3-acetylation was detected upon DCAF15 degradation within the same cell cycle phase. As such, we tested whether aberrant accumulation of cohesin-bound CDCA5/PDS5A in the previous cell cycle phase would result in defective cohesin acetylation in the immediately subsequent cell cycle. To this purpose, DCAF15 was degraded by dTAG-13 in G2-synchronized cells. Upon G2 release and progression into the next cell cycle, cells lacking DCAF15 displayed an impairment in sustaining high levels of acetylated-SMC3 (Fig. 5d, Supplementary Fig. 5c).

To assess the extent of the SMC3-acetylation defect on chromatin upon *DCAF15* loss, we performed chromatin immunoprecipitation followed by sequencing (ChIP-seq) analysis of total-SMC3 and acetylated-SMC3. Initial analysis of significant peaks distribution across genome regions did not reveal changes in the distribution of total- or acetylated-SMC3 upon *DCAF15*-knockout (Supplementary Fig. 5d and Supplementary Data 4). Notably, MEME-ChIP[53] analysis confirmed that total- and acetylated-SMC3 peaks retrieved almost exclusively overlapped with CTCF peaks (Supplementary Fig. 5e), as has been well-established[54,55]. However, a large majority of acetylated-SMC3 significant peaks in *DCAF15*$^{+/+}$ cells were not scored in *DCAF15*$^{-/-}$ cells (Fig. 5e). In addition to peaks-calling, fold change-based analysis of total- and acetylated-SMC3 peaks also confirmed a genome-wide defect of DNA-bound acetylated-SMC3 upon *DCAF15* loss (Fig. 5f, g, Supplementary Fig. 5f). Importantly, the same analysis did not show a defect in total-SMC3 loading onto DNA, suggesting that the cohesin complex is properly loaded onto DNA but not properly acetylated in *DCAF15*$^{-/-}$ cells. Targeted ESCO1 ChIP-qPCR at genomic sites revealed diminished ESCO1 binding in *DCAF15*$^{-/-}$ cells (Supplementary Fig. 5g), suggesting a defect of ESCO1 cohesin acetyltransferase recruitment on cohesin upon DCAF15 loss.

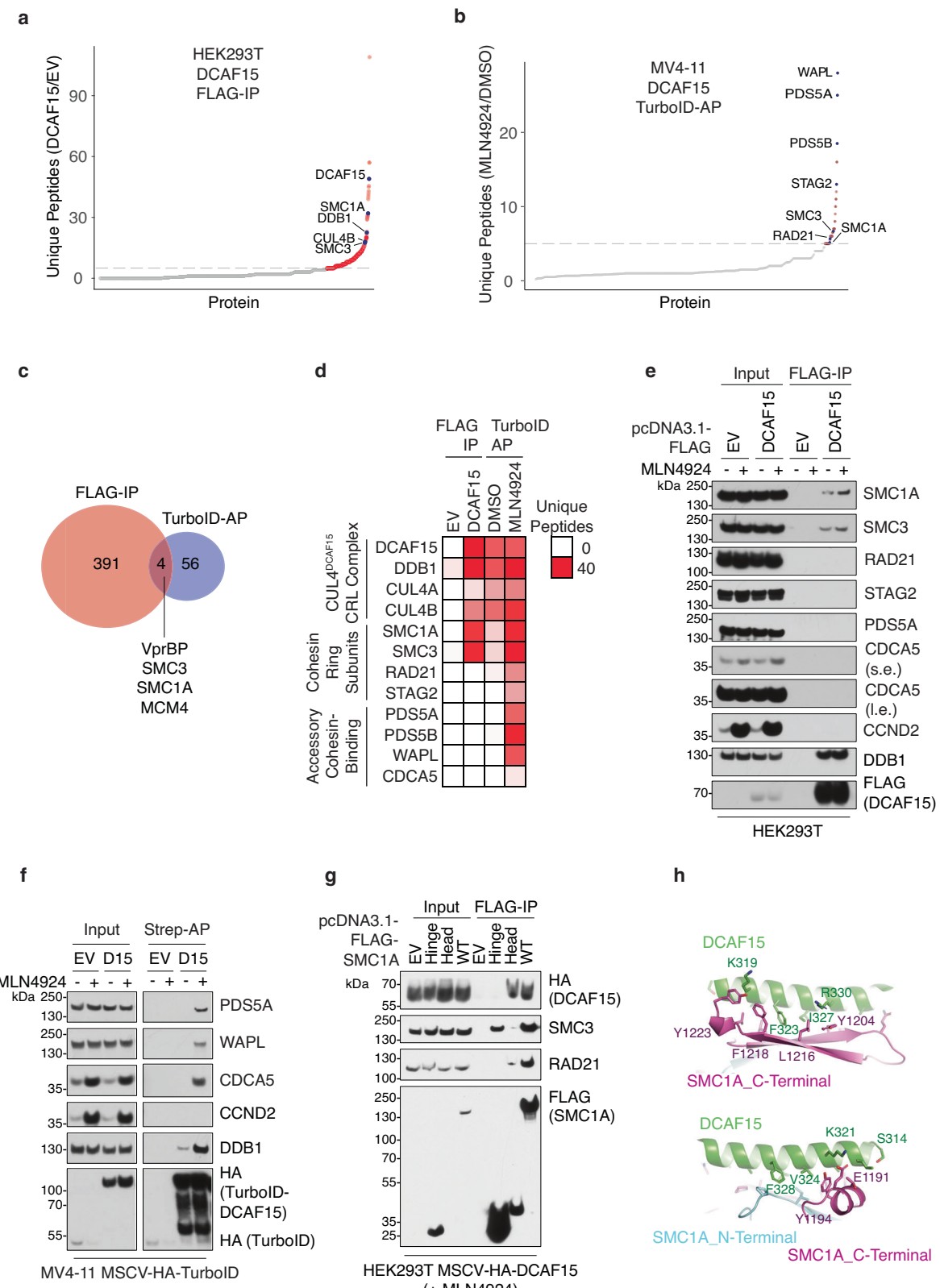

Taken together, these results suggested that DCAF15 binds to cohesin that has been de-acetylated by HDAC8 and promotes degradation of PDS5A (and indirectly CDCA5) in G2 and prometaphase. This process contributes to refreshing cohesin for ESCO1/2-mediated re-acetylation upon cohesin re-loading onto chromatin in the following cell cycle.

## DCAF15 loss results in enlargement of chromatin loops

Deficient cohesin acetylation on chromatin has been shown to cause defective translocation of cohesin on DNA[33,56,57] and dysregulated DNA loop extrusion[33,34,52,58–60]. Therefore, we sought to determine whether *DCAF15* loss results in changes in DNA looping and the formation of topologically associating domains (TADs). To this end, we conducted

**Fig. 3 | DCAF15 interacts with the cohesin complex. a** Mass spectrometry analysis of FLAG empty vector (EV) or FLAG-DCAF15 immunoprecipitants in HEK293T cells. Scatter plot shows the ratio between the unique peptides in the FLAG-DCAF15 versus EV samples. Red and blue circles represent DCAF15 interactors enriched by at least 5-fold compared to EV. **b** Mass spectrometry analysis of biotin-streptavidin (Strep)-affinity purification (AP) from MV4-11 cells stably expressing HA-tagged TurboID-DCAF15 treated with DMSO or 5 μM MLN4924 for 6 h; biotin was added at 50 μM for final 2 h. Scatter plot shows the ratio between the unique peptides in the MLN4924 versus DMSO conditions. Red and blue circles represent the proteins enriched by at least 5-fold in the presence of MLN4924 compared to DMSO. **c** Venn diagram showing the overlap of DCAF15 FLAG-interactome (**a**) and proximity-interactome (**b**). **d** Heatmap showing number of unique peptides identified in (**a**) and (**b**) for the CUL4^DCAF15 CRL complex, core cohesin ring subunits, and accessory cohesin-binding proteins. **e** Lysates from HEK293T cells transiently transfected with FLAG EV or DCAF15, treated with DMSO or 5 μM MLN4924 for final 6 h, were subjected to FLAG-immunoprecipitation and Western blot analysis for the indicated proteins. Immunoblots are representative of three independent experiments. **f** Lysates from MV4-11 HA-TurboID-EV and HA-TurboID-DCAF15 (D15) cells treated with DMSO or 5 μM MLN4924 for 6 h and 50 μM biotin for final 2 h were subjected to biotin-streptavidin (Strep)-affinity purification (AP) and Western blot analysis for the indicated proteins. Immunoblots are representative of three independent experiments. **g** Lysates from HEK293T cells stably expressing HA-tagged DCAF15 and transiently transfected with FLAG EV, FLAG-SMC1A-Hinge-Domain-Only (Hinge), FLAG-SMC1A-Head-Domain-Only (Head), or FLAG-SMC1A-WT, treated with 5 μM MLN4924 for 6 h, were subjected to FLAG-immunoprecipitation and Western blot analysis for the indicated proteins. Immunoblots are representative of three independent experiments. **h** AlphaFold[44] prediction analysis in ChimeraX[89] of DCAF15 (green), SMC1A_C-Terminal (magenta), and SMC1A_N-Terminal (cyan). Source data are provided as a Source Data file.

Hi-C chromosome conformation capture on control and *DCAF15*-knockout cells. Strikingly, loss of *DCAF15* resulted in genome-wide loss of close-range DNA contacts concomitant with gain of longer-range contacts (Fig. 6a), indicative of uncontrolled loop extrusion. Consistent with this observation, visual inspection of Hi-C matrices revealed that in *DCAF15*-knockout cells, some adjacent TADs coalesce to form larger TADs, indicating weakening of boundary strength (Fig. 6b; Supplementary Fig. 6a shows widespread differences in large portion of whole chromosome). Metaplots of TADs confirmed loss of interactions within TADs concomitant with gain of inter-TAD contacts upon *DCAF15* loss (Fig. 6c). Interestingly, sites of lost TAD boundaries often aligned with sites of diminished acetylated-SMC3 chromatin occupancy in cells lacking *DCAF15* (Fig. 6b), suggesting that loss of cohesin acetylation may result in dysregulated maintenance of 3D genome architecture.

Next, we assessed the effect of *DCAF15* loss on DNA looping interactions. We found that the median loop length in *DCAF15*^−/− cells was significantly increased compared to control cells (Fig. 6d), consistent with a genome-wide gain in long-range contacts. Aggregate peak analysis (APA) showed that in cells lacking *DCAF15*, larger-sized loops were strengthened, while smaller loops showed weakening of contacts (Fig. 6e). Visualization of looping interactions in the control versus *DCAF15* loss condition confirmed clear increases in contacts over larger loops that are gained upon DCAF15-depletion (Fig. 6f, arrows and red arcs). Similar to our observation with dissolving TAD boundaries, sites of *DCAF15*-specific loop extrusion are often aligned with sites of diminished acetylated-SMC3 chromatin occupancy, suggesting that DCAF15-mediated sustaining of cohesin acetylation may be a general mechanism required for proper maintenance of DNA contacts of various sizes.

Taken together, these results demonstrated that DCAF15 regulation of cohesin acetylation is necessary for maintaining proper genome architecture.

### DCAF15 loss disrupts DNA replication fork integrity

Dysregulated DNA loop extrusion has been connected to defective DNA replication fork progression and increased susceptibility to DNA damage[23,61,62]. Accordingly, we assessed the integrity of DNA replication fork progression upon *DCAF15* loss. As no major changes in cell cycle distribution were detectable at steady state (Supplementary Fig. 7a), a DNA fiber assay was performed to calculate replication fork recovery following hydroxyurea (HU)-induced replication fork stalling. In order to quantify fork restart efficiency after release from HU-induced replication stalling, we calculated relative IdU/CldU tract length ratios (Fig. 7a). Compared to control cells, *DCAF15*-knockout cells displayed a significant reduction in median IdU/CldU ratio, revealing compromised replication recovery (Fig. 7b).

Next, we determined whether *DCAF15* loss led to accumulation of DNA damage in AML. Indeed, *DCAF15*-knockout cells displayed accumulation of γH2AX, a marker of DNA double-strand breaks, independently of p53 activation (Fig. 7c, Supplementary Fig. 7b, c), suggesting that p53 activation in *DCAF15*-knockout cells is downstream of DNA damage response activation. Following, we questioned whether *TP53*-WT AML cells lacking *DCAF15* may be more susceptible to replication-dependent DNA damage induced by DNA-damaging therapeutics. Interestingly, a chemogenetic screen performed in HAP-1 cells suggested that *DCAF15* loss renders cells sensitive to the topoisomerase I inhibitor camptothecin[63]. A competition-based proliferation assay confirmed that *DCAF15*-knockout MV4-11 cells were outcompeted by parental cells at a faster rate in the presence of CPT compared to DMSO, while cells transduced with a non-targeting sgRNA were not (Fig. 7d). Furthermore, knockout of *TP53* rescued the anti-proliferative and pro-apoptotic effects in CPT-treated *DCAF15*-knockout cells (Supplementary Fig. 7d, e), suggesting that the increased CPT-sensitivity of MV4-11 cells upon *DCAF15*-knockout is due to activation of the p53. Similar data were observed when cells were treated with olaparib, a PARP inhibitor that induces replication-dependent DNA damage and cytotoxicity by trapping the PARP DNA repair enzyme on DNA[64] (Fig. 7e).

These results suggested that DCAF15 maintains replication fork integrity to prevent accumulation of DNA damage and suppress p53 activation. Altogether, our findings reveal a function of DCAF15 in controlling cohesin complex recycling and acetylation dynamics to maintain proper DNA loop extrusion, preserve replication fork integrity, and sustain cell proliferation (Fig. 7f).

## Discussion

Cohesin is vital throughout the cell cycle through its role in 3D genome organization[33], and the cohesin acetylation cycle dynamically controls the length of extruded DNA loops[52]. More specifically, ESCO1-mediated SMC3-acetylation pauses chromatin loop extrusion, and conversely, HDAC8-mediated SMC3-deacetylation alleviates this braking mechanism to promote loop enlargement[52]. In our present study we showed that, in *DCAF15*-knockout cells, cohesin localized to CTCF chromatin loop extrusion sites is defective in acetylation of SMC3. Consistent with depleted SMC3-acetylation, our chromosome conformation capture-based studies have demonstrated genome-wide enlargement of chromatin loops upon *DCAF15* loss. We also revealed that *DCAF15*-deficient cells accumulate cohesin hyper-loaded with PDS5A/CDCA5, and hypo-loaded with ESCO1. Thus, we propose that cohesin-bound PDS5A/CDCA5 sterically prevents ESCO1/2-mediated SMC3-acetylation, however further structural mechanistic details of precisely how this is achieved remain to be elucidated.

Regulation of cohesin dynamics is critically important in conditions of DNA replication stress[65]. ESCO1/2-mediated acetylation of SMC3 in front of the replication fork is required for processive movement past the cohesin complex[56], and behind the fork, cohesin acetylation is promoted by DNA flap and nick structures that

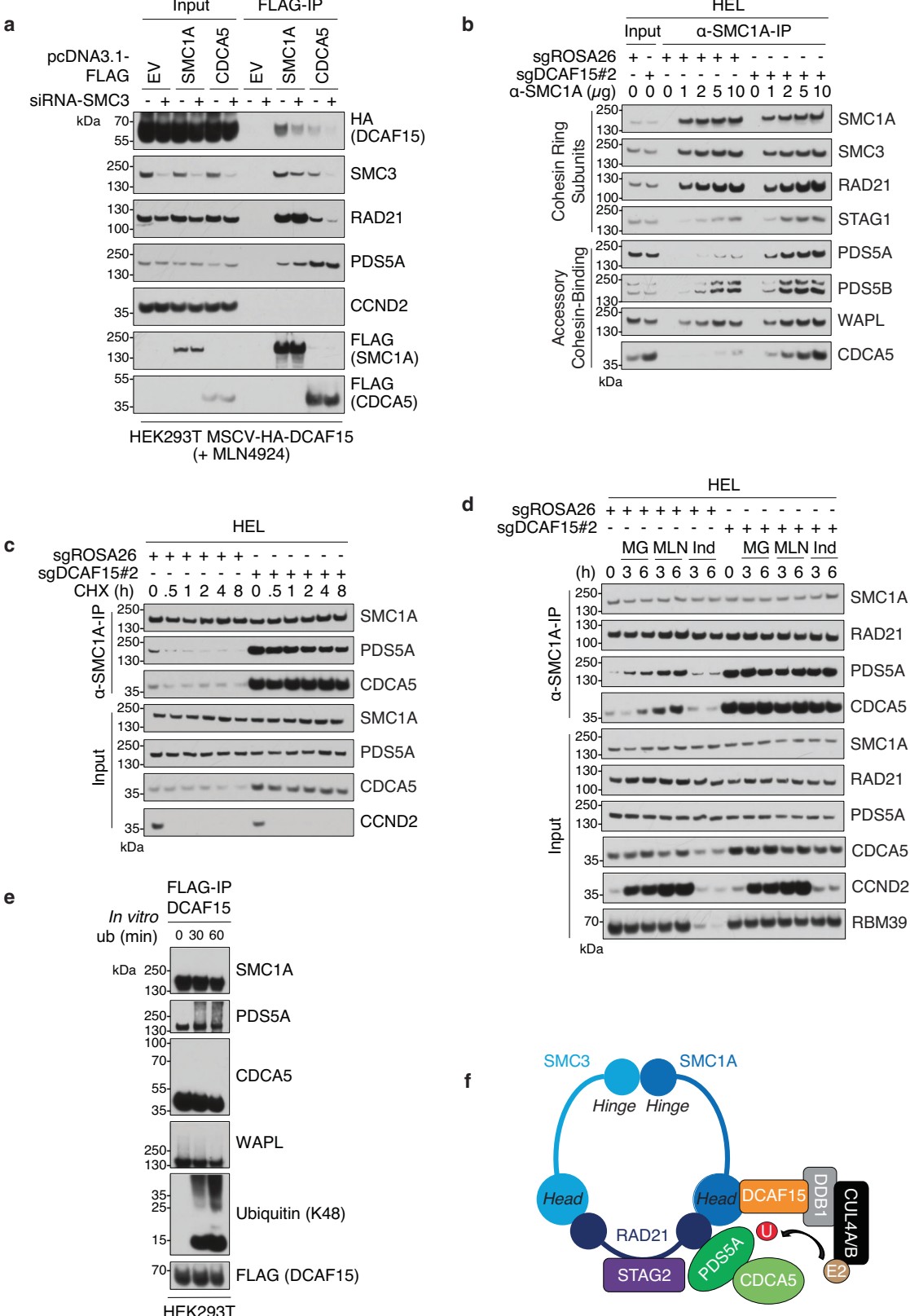

characterize Okazaki fragment maturation to stabilize newly established sister chromatid cohesion[66]. Here, we showed that loss of *DCAF15* leads to replication stress and accumulation of DNA damage, resulting in the induction of p53-dependent apoptosis. As such, it is intriguing to speculate that the expansion of chromatin loops in *DCAF15*-knockout cells may be causative of defective DNA replication

fork progression. Indeed, *DCAF15*-knockout cells were highly sensitive to topoisomerase I-inhibition, suggesting a topological defect of DNA supercoiling under these conditions.

Our biochemical and structural analyses revealed that a conserved DCAF15 α-helix makes direct contacts to both the N-terminal and C-terminal domains that form the head domain of SMC1A. Of note,

**Fig. 4 | DCAF15 destabilizes cohesin-bound PDS5A and CDCA5. a** Lysates from HEK293T cells stably expressing HA-tagged DCAF15 with the indicated siRNA and/or FLAG-tagged cDNAs, treated with 5 μM MLN4924 for 6 h, were subjected to FLAG-immunoprecipitation and Western blot analysis for the indicated proteins. Immunoblots are representative of two independent experiments. **b** Lysates from Cas9+ HEL cells infected with lentiviruses encoding *ROSA26-* or *DCAF15*-targeting sgRNAs were subjected to immunoprecipitation with the indicated amounts of α-SMC1A antibody and Western blot analysis for the indicated proteins. Immunoblots are representative of three independent experiments. **c** Same as in (**b**), except that cells were treated with 5 μg/mL cycloheximide (CHX) for indicated times. Immunoblots are representative of two independent experiments. **d** Same as in (**b**),

except that cells were treated with 10 μM MG132 (proteasome inhibitor, MG), 5 μM MLN4924 (MLN), or 3 μM indisulam (Ind) for indicated times. Immunoblots are representative of three independent experiments. **e** Lysates from HEK293T cells transiently transfected with FLAG-DCAF15, HA-SMC1A, HA-PDS5A, and HA-CDCA5 were subjected to FLAG-immunoprecipitation. Following, FLAG-immunoprecipitants were subjected to in vitro ubiquitination assay and Western blot analysis for the indicated proteins. Immunoblots are representative of two independent experiments. **f** Model of CRL4-DCAF15 binding the head domain of SMC1A and promoting the ubiquitination of PDS5A when bound to the SMC1A-SMC3-RAD21 tripartite ring. Source data are provided as a Source Data file.

our structural analysis of the mode of DCAF15-SMC1A interaction revealed that the internal loop α-helix of DCAF15 and the C-terminal domain of RAD21 bind at the same site on SMC1A. Therefore, while one could speculate that DCAF15 physically competes with RAD21 for binding to SMC1A, RAD21 makes additional contacts with the rest of the cohesin complex, including interfaces with SMC3 and STAG1/2[45]. Thus, the α-helix of DCAF15 does not necessarily dislodge RAD21 from the cohesin complex entirely; instead, the E3 may simply remodel the intact ring-shaped assembly. These observations could help to interpret our finding that knocking down *RAD21* weakens the DCAF15-SMC1A interaction, suggesting that DCAF15 co-exists with RAD21 on SMC1A.

The cohesin-DCAF15 relationship is of particular relevance in the context of acute myeloid leukemia (AML)[67,68]. Compared to other cancer types, mutations in p53 are much less frequent in AML, occurring in approximately 5% of de novo cases[69]. Importantly, other genes that are recurrently mutated in AML include those composing the cohesin complex (*SMC1A, SMC3, RAD21, STAG2*), together associated with more than 10% of cases[21,22]. A common standard-of-care regimen for newly diagnosed AML patients is combination of an anthracycline (DNA-damaging agent) with a nucleoside analog (such as cytarabine)[70,71]. While initial remission rates are relatively high, complete remission is rare as most patients develop drug resistance, and disease relapse portends a dire outcome[72]. Therefore, additional treatment regimens to overcome relapsed disease are of great necessity. Our CRISPR-based functional genetic screens identified *DCAF15* as an AML-biased vulnerability. In AML, high *DCAF15* expression correlates with worse overall patient survival prognosis. Additionally, *DCAF15* is up-regulated in AML patients compared to normal hematopoietic cells, suggesting that leukemia cells may hijack normal DCAF15 function during oncogenesis. Importantly, we have shown that *DCAF15* loss sensitizes AML to the topoisomerase I inhibitor camptothecin as well as the PARP inhibitor olaparib. This suggests that the endogenous replication-dependent DNA damage and cytotoxicity occurring in cells lacking *DCAF15* can be exacerbated with exogenous DNA-damaging therapeutics. As such, approaches to target DCAF15 in conjunction with DNA-damaging therapies could result in beneficial treatment avenues for AML.

Of note, a genome-wide CRISPR screen in AML cells found that ablation of *DCAF15* sensitizes cells to venetoclax (a clinical BCL2 inhibitor)-induced apoptosis[73]. Additionally, a CRISPR screen performed in chronic myeloid leukemia (CML) cells demonstrated that DCAF15 is a negative regulator of natural killer (NK) cell-mediated clearance of cancer cells[74]. In this report, the cohesin subunits SMC1A and SMC3 were proposed as endogenous substrates of DCAF15, however the significance of these observed interactions was not further explored.

In summary, we have identified *DCAF15* as an AML-biased vulnerability. DCAF15 controls cohesin complex recycling dynamics through specific binding to SMC1A and destabilization of the cohesin-bound regulatory factors PDS5A and CDCA5. Importantly, we propose that DCAF15-mediated removal of PDS5A/CDCA5 from cohesin is required to enable acetylation of cohesin on chromatin. Loss of

DCAF15-mediated cohesin complex recycling leads to compromised replication fork integrity and dysregulated DNA loop extrusion. Critically, *DCAF15* loss sensitizes AML to replication stress-inducing therapeutics, through activation of p53-dependent apoptosis. Altogether, this study reveals and details the endogenous and cell autonomous function of DCAF15, as a key regulator of cell proliferation via control of cohesin dynamics.

## Methods
### Cell culture
THP-1 (ATCC, TIB-202, male), SET-2 (DSMZ, ACC-608, female), HEL (ATCC, TIB-180, male), U-937 (ATCC, CRL-1593.2, male), MOLM-13 (DSMZ, ACC-554, male), OCI-AML3 (DSMZ, ACC-582, male), MV4-11 (ATCC, CRL-9591, male), Jurkat (ATCC, TIB-152, male), U-2932 (DSMZ, ACC-633, female), and OPM-1 (DSMZ, ACC-50, female) cells were maintained in RPMI-1640 media (Gibco) supplemented with 10% fetal bovine serum (FBS) (Hyclone) and 1% Penicillin/Streptomycin. Hep-G2 (ATCC, HB-8065, male) cells were maintained in DMEM media (Corning) supplemented with 10% FBS (Hyclone) and 1% Penicillin/Streptomycin. HEK293 (ATCC, CRL-1573, female) and HEK293T (ATCC, CRL-3216, female) cells were maintained in DMEM media (Corning) supplemented with 10% bovine serum (BS) (Gibco) and 1% Penicillin/Streptomycin. Cells were incubated at 37 °C with 5% $CO_2$.

### sgRNA and plasmid cloning
Cancer cell lines expressing Cas9 were constructed through lentiviral transduction of an spCas9 expression vector (Addgene, 108100) followed by selection with puromycin (Sigma, P8833). sgRNAs were cloned by annealing sense and antisense DNA oligos prior to ligation into Esp3I (BsmBI)-digested LRG2.1 (Addgene, 108098) or LRCherry2.1 (Addgene, 108099) plasmid backbones. sgRNA sequences used are listed in Supplementary Data 1.

For transient expression experiments, full-length human *DCAF15* cDNA was PCR-amplified and introduced into a pcDNA3.1 vector containing an N-terminal 2X-FLAG-tag and 2X-Strep-tag by restriction enzyme cloning (Thermo Scientific FastDigest system). Truncated and mutated forms of DCAF15 were also derived from this plasmid through PCR-mutagenesis. Human SMC1A transient expression vectors were similarly constructed after PCR-amplification of cDNA from the pcDNA3 5′ cMyc SMC1 wt vector (Addgene, 32363). The pcDNA3.1 C-terminal FLAG-tagged full-length human CDCA5 vector was purchased from GenScript (OHu11344).

For construction of *DCAF15* stable viral expression vectors, full-length *DCAF15* cDNA was PCR-amplified with an additional N-terminal HA-tag from the aforementioned pcDNA3.1 vector and cloned into the MSCV-Puro retroviral vector (Addgene, 68469), with an additional N-terminal 3X-HA-tag-TurboID cDNA cloned in, and the pBABE-GFP retroviral vector (Addgene, 10668) using restriction enzyme cloning. The CRISPR-resistant synonymous substitution of nucleotide G207 to C was introduced into the *DCAF15* cDNA with PCR-mutagenesis. The TRIPZ doxycycline-inducible lentiviral human *DCAF15* shRNA vector was purchased from Horizon Discovery (RHS4740-EG90379).

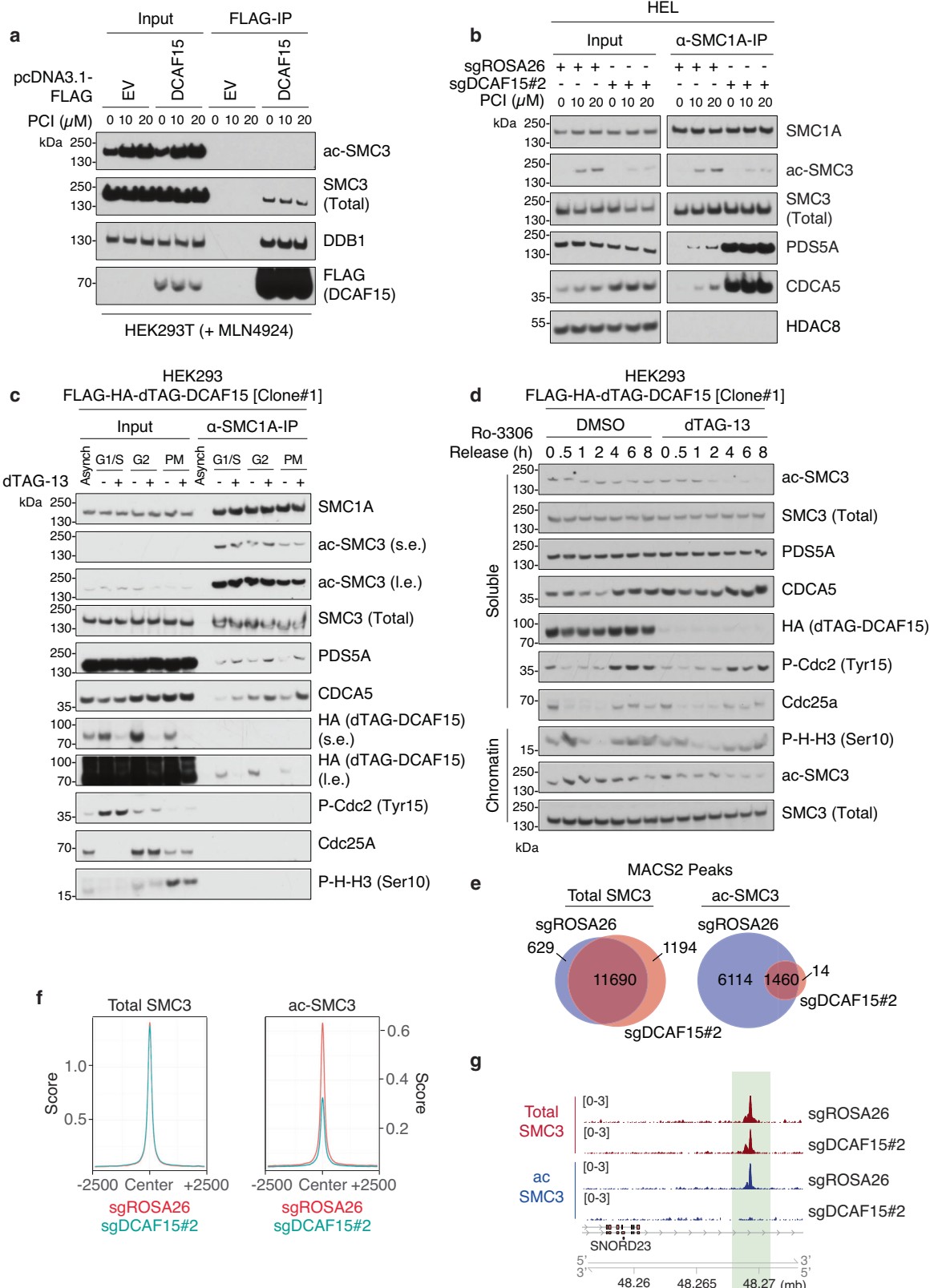

### Generation of FLAG-HA-FKBP12(F36V) knock-in cell lines

Single-guide RNA (sgRNA) targeting ± 60 bp around the start codon of the *DCAF15* gene was designed using web-based CRISPR design tool (https://benchling.com/). The homology repair fragment spanning the *DCAF15* start codon (450 bp on each side) and containing the FLAG-HA-FKBP12(F36V)-P2A-GFP protein-coding gene and linker sequence

(dTAG cassette[40]) was designed with the corresponding PAM site mutated, which was then cloned into a pUC57 construct obtained from Genewiz. Following this, Cas9 protein V2, synthetic guide RNA, and FLAG-HA-FKBP12(F36V)-knock-in (KI) donor template were co-transfected into cells using the Neon nucleofector. GFP-high cells were sorted 72 h post-transfection using FACS, and the sorted cells

**Fig. 5 | DCAF15 sustains cohesin acetylation on chromatin. a** Lysates from HEK293T cells transiently transfected with FLAG EV or FLAG-DCAF15 treated with DMSO or the indicated concentrations of PCI-34051 (HDAC8 inhibitor, PCI), and treated with 5 μM MLN4924 for 6 h, were subjected to FLAG-immunoprecipitation and Western blot analysis for the indicated proteins. Immunoblots are representative of three independent experiments. **b** Lysates from Cas9+ HEL cells infected with lentiviruses encoding *ROSA26*- or *DCAF15*-targeting sgRNAs, treated with DMSO or the indicated concentrations of PCI, were subjected to immuno-precipitation with α-SMC1A antibody and Western blot analysis for the indicated proteins. Immunoblots are representative of three independent experiments. **c** HEK293 FLAG-HA-dTAG-DCAF15[Clone#1] cells were synchronized by double thymidine block (G1/S) followed by Ro-3306 block (G2) or nocodazole block (PM prometaphase), or left asynchronous (Asynch), and treated with DMSO or dTAG-13 (100 nM) for final 6 h prior to harvest. Lysates were subjected to immunoprecipitation with α-SMC1A antibody and Western blot analysis for the indicated proteins. Immunoblots are representative of two independent experiments. **d** HEK293 FLAG-HA-dTAG-DCAF15[Clone#1] cells were synchronized by double thymidine block followed by Ro-3306 block (3 μM for 24 h) and treated with DMSO or dTAG-13 (100 nM) for final 6 h of block. Following, cells were washed with PBS and released in fresh media containing DMSO or dTAG-13 (100 nM). Cells were harvested at indicated time points, and cell lysates from the cytoplasmic (Soluble) and nuclear (Chromatin) fractions were analyzed by Western blot for the indicated proteins. Immunoblots are representative of two independent experiments. **e** Venn diagrams showing the overlap of ChIP-seq peak counts for Total-SMC3 (Left) and ac-SMC3 (Right) in Cas9+ HEL cells infected with lentiviruses containing *ROSA26*- or *DCAF15*-targeting sgRNAs. **f** Enrichment plots displaying the average ChIP-seq signals of Total-SMC3 (Left) or ac-SMC3 (Right) in Cas9+ HEL cells infected with lentiviruses containing *ROSA26*- or *DCAF15*-targeting sgRNAs. 2500 bp upstream and down-stream of peak centers are shown. **g** Representative genome browser track (produced using UCSC Genome Browser) showing Total-SMC3 (Top) and ac-SMC3 (Bottom) levels in Cas9+ HEL cells infected with lentiviruses containing *ROSA26*- or *DCAF15*-targeting sgRNAs. Source data are provided as a Source Data file.

were cultured and expanded for an additional week. Single-cell clones were subsequently plated into 96-well plates via FACS. By day 15, small colonies of clonal cells were visible, and these cells were screened by PCR to determine homozygosity.

### Virus production and transduction

For the production of lentiviruses, HEK293T cells were transfected with the lentiviral plasmid of interest, lentiviral packaging plasmids (pPAX2 and VSVG), and polyethylenimine (PEI; 1 mg/mL). Lentiviruses were produced in 10-cm plates seeded at 90% cellular confluency, and comprised 5 μg plasmid of interest, 3.75 μg pPAX2, 2.5 μg, VSVG, 80 μL PEI, and 1 mL OPTI-MEM serum-free media. Cells were incubated with transfection reagents for ~6 h prior to removing and refreshing media. Lentivirus-containing media was harvested and pooled at 24 h, 48 h, and 72 h post-transfection.

For the production of retroviruses, HEK293T cells were trans-fected with the retroviral plasmid of interest, retroviral packaging plasmids (GP and VSVG), and polyethylenimine (PEI; 1 mg/mL). Ret-roviruses were produced in 10-cm plates seeded at 90% cellular con-fluency, and comprised 4 μg plasmid of interest, 2 μg GP, 2 μg, VSVG, 56 μL PEI, and 500 μL 150 mM NaCl diluent solution. Lentivirus-containing media was harvested and pooled at 24 h and 48 h post-transfection.

For transduction of viruses, 0.45 μM membrane-filtered virus-containing media and polybrene (2 mg/mL) were added to cells plated at 150,000 cells/mL in 24-well plates. Cells were spin-infected for 25 min at 609 × g at room temperature. Cells were incubated with virus transduction reagents for ~3 h prior to removing and refreshing with virus-free media.

### siRNA transfection

HEK293T cells seeded at ~50% confluency in 10-cm plates were trans-fected with 600 pmol siRNA and 30 μL Lipofectamine-2000 (Thermo Fisher Scientific, 11668027) for 48 h. Cells were incubated with siRNA transfection reagents for 6 h prior to removing and refreshing the media. The following siRNAs were used: ON-TARGETplus Human SMC3 (9126) siRNA SMARTPool (Horizon Discovery, L-006834-00-0005).

### Domain-focused CRISPR screen

Performed as in Zhou et al. [35]. The human Cullin-RING E3 ubiquitin ligase (CRL) domain-focused CRISPR sgRNA library was designed according to conserved domain annotation information available from the NCBI database. At least five to six independent sgRNAs were designed to target each gene. In addition to 20 positive control sgRNAs targeting 6 well known cell-essential genes (*CDK1*, *CDK9*, *PCNA*, *POLR2A*, *RPA3*, *RPL23A*) and 100 negative control non-targeting sgRNAs, the final library contained 2405 sgRNAs targeting 381 CRL family genes. Pooled sgRNAs were synthesized (Twist Bioscience) and

cloned into Esp3I (BsmBI)-digested LRCherry2.1 (Addgene, 108099) plasmid backbone via Gibson cloning (NEB). Representation and identity of sgRNA library were verified with deep sequencing analysis.

As detailed above, spCas9+ cells were constructed via lentiviral transduction of the Lenti_Cas9_Puro vector (Addgene, 108100) and puromycin-selected to 100% positivity prior to sgRNA library transduction.

Pooled library was delivered by lentiviral transduction at a mul-tiplicity of infection (MOI) between 0.3 and 0.5 (confirmed via mea-suring of mCherry+ % on day 3 post-infection), to ensure that individual cells obtained only one sgRNA copy upon transduction. Cells were grown for 5 population doubling times and split as required while maintaining at least 1000X representation of each sgRNA. For an initial timepoint, cells were harvested on day 3 post-infection, for comparison of sgRNA abundance to cells harvested for a final time-point 5 population doubling times later. Cell pellets containing mCherry+ and mCherry- cells were washed with PBS and frozen at −80 °C until extraction of genomic DNA. Genomic DNA was extracted with the Quick-DNA Miniprep Kit (ZYMO), according to the manu-facturer's protocol. DNA was eluted with molecular-grade PCR water and frozen at −20 °C until sequencing library preparation.

Sequencing library was prepared by PCR-amplifying the inte-grated sgRNA cassette from genomic DNA (~300 ng input) with cus-tom stacking barcode incorporation[75]. Each library was amplified with a different barcode to ~100 ng final PCR product. Gel-extracted pro-ducts were purified twice using the Macherey-Nagel NucleoSpin Gel and PCR Clean-Up mini kit, and eluted with molecular-grade PCR water. Illumina sequencing adaptors were then added to the barcode-embedded products with 8 cycles of PCR-amplification, and final PCR products were purified using the QIAquick PCR purification kit and eluted with 30 μL PCR water (QIAGEN). Libraries were analyzed for target product size (~320 bp) and high quality using the Bio-analyzer DNA 1000 kit (Agilent). Concentration of library was calculated with the Qubit dsDNA HS assay kit (Thermo Fisher). Libraries with different barcodes were pooled to 4 nM using the online Illumina pooling cal-culator. 4 nM pooled library was then denatured to 20 pM according to the Illumina protocol, and 600 μL of the 20 pM pool was loaded into the cartridge. Libraries were sequenced on the MiSeq or NextSeq 500 platforms with 75 bp single- or paired-end reads.

Sequencing reads were de-multiplexed and trimmed to preserve only the sgRNA cassette. Data were aligned to the reference sgRNA library with no mismatch tolerated[36]. All samples were normalized to the same number of total reads, and the average $\log_2$(Fold Change) of the sgRNA abundance of each gene (refined as essentiality score, ES) was determined[76]. AML-biased ES was computed by subtracting the average ES of the AML cell lines (MOLM-13 and MV4-11) from the average ES of the other cell lines (Jurkat, U-2932, OPM-1, Hep-G2). The human Cullin-RING E3 ubiquitin ligase (CRL) domain-focused CRISPR

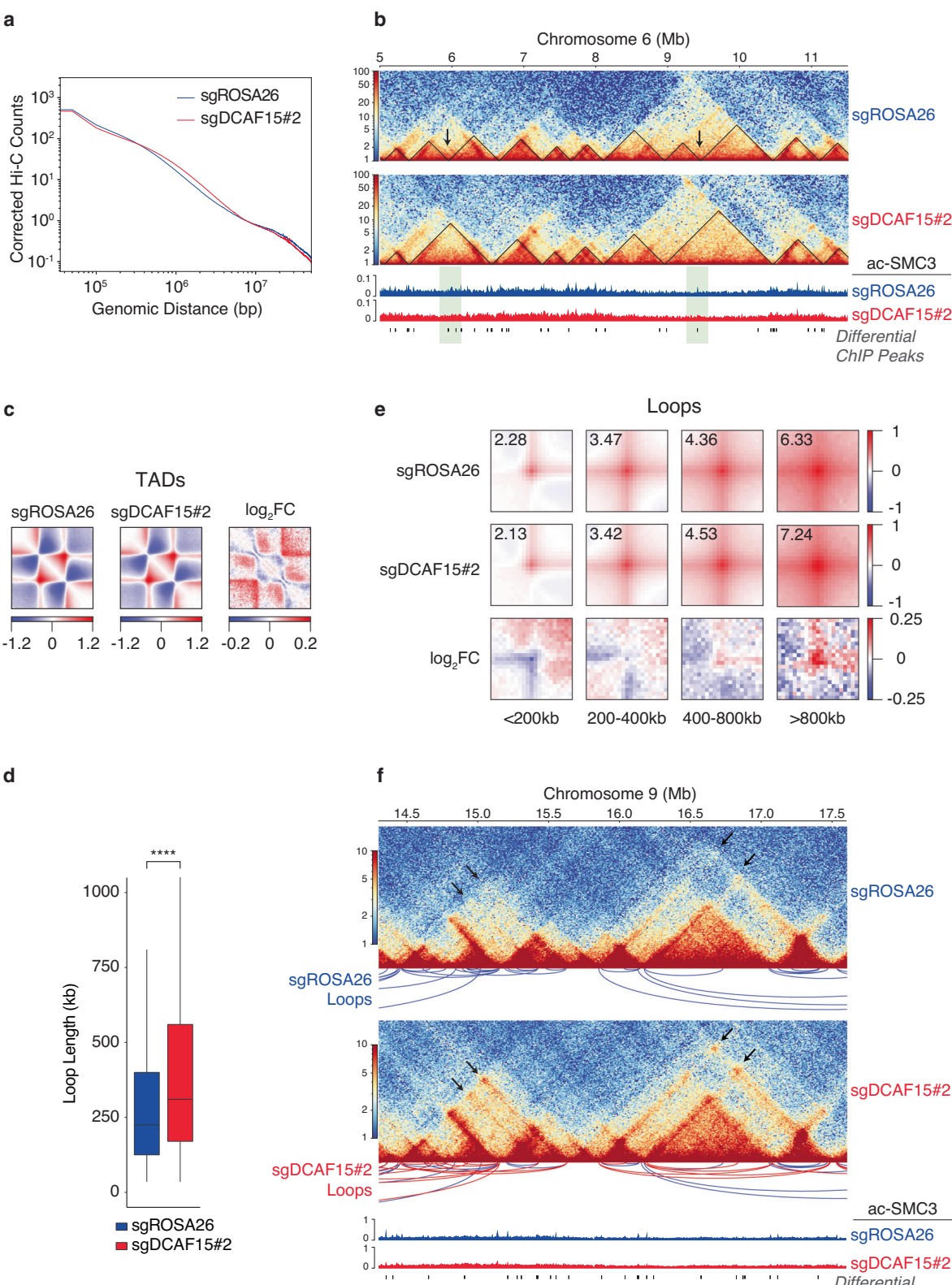

screen data for all 6 cancer cell lines is available in Supplementary Data 1.

## Competition-based cell proliferation assay

For individual gene knockout validation, Cas9+ cell lines were transduced with LRG2.1 or LRCherry2.1 sgRNA-containing lentiviral vectors co-expressed with a GFP or mCherry reporter, respectively. Percentage of fluorescence-positive cells corresponded to the sgRNA representation within the mixed population. Using an Attune NxT flow cytometer (Thermo Fisher), % fluorescence readings were acquired beginning on day 3 post-infection, and every 3–4 days after for up to 25 days post-infection. Flow cytometry data was processed using FlowJo software, and relative fluorescence proportion (normalized to day 3 post-infection) was used for analysis.

**Fig. 6 | *DCAF15* loss results in enlargement of chromatin loops. a** Contact probability as a function of genomic distance in Cas9+ HEL cells infected with lentiviruses encoding *ROSA26*-targeting (negative control) or *DCAF15*-targeting sgRNAs. Hi-C data derived from *n* = 2 independent biological replicates in each condition. **b** Representative region in chromosome 6 (chr6:05,000,000–11,500,000) showing contacts, distribution of ac-SMC3, and sites with differential acetylation of SMC3. Black boundaries delineate TADs called in each condition. Arrows indicate domain boundaries lost in *DCAF15*−/− cells. **c** 3D pileup plots of TADs in each condition and log$_2$(Fold Change) upon loss of *DCAF15*. **d** Box plot showing length distributions of loops called in *DCAF15*+/+ versus *DCAF15*−/− cells. *n* = 10,662 loops (sgROSA26) and 11,871 loops (sgDCAF15#2). Center line shows median, box limits show 75th and 25th percentiles, whiskers show 1.5X interquartile range (IQR); outliers not shown. sgROSA26 median loop length: 225 kb, sgDCAF15#2 median loop length: 310 kb. Welch's two-sided t-test, ****p-value < 0.0001 ($p = 2.2 \times 10^{-16}$). **e** 3D pileup plots of loops categorized by indicated size ranges in each condition and log$_2$(Fold Change) upon *DCAF15* loss. **f** Representative region in chromosome 9 (chr9:14,000,000–18,000,000) showing contacts, distribution of ac-SMC3, and sites with differential acetylation of SMC3. Loops present in control only or in both conditions are indicated in blue. Loops gained in *DCAF15*-knockout condition only are indicated in red. Arrows indicate example regions of increased contact in *DCAF15*−/− cells.

For double gene knockouts, Cas9+ cell lines were transduced with LRG2.1 and LRCherry2.1 sgRNA-containing lentiviral vectors simultaneously, or in succession. % fluorescence readings from the mixed populations (uninfected, GFP+, mCherry+, and GFP+/mCherry+) were acquired beginning on day 3 post-infection and every 3–4 days after for up to 25 days post-infection. Relative fluorescence proportions (normalized to day 3 post-infection) were used for analysis.

### Drug sensitivity assays
To assay sensitization to camptothecin (CPT) drug by gene knockout, competition-based cell proliferation assays were carried out as detailed above, with cells grown in the presence of drug. Cas9 + MV4-11 cells were plated in 24-well plates at 50,000 cells/mL, and CPT (Sigma–Aldrich, C9911-100MG) was added to culture media at 1 nM. 150 μL of cells were collected at 1, 2, 3, and 4 days post-drug treatment, for flow cytometric analysis of relative fluorescence-positive (sgRNA+) proportions (normalized to treatment with DMSO vehicle control).

To assay sensitization to the olaparib (Ola) drug by gene knockout, competition-based cell proliferation assays were carried out as detailed above, with cells grown in the presence of drug. Cas9 + MV4-11 cells were plated in 24-well plates at 50,000 cells/mL, and olaparib (LC Laboratories, O-9201) was added to culture media at 0.5 μM and 1 μM, respectively. 150 μL of cells were collected at 7 days post-drug treatment, for flow cytometric analysis of relative fluorescence-positive (sgRNA+) proportions (normalized to treatment with DMSO vehicle control).

### Apoptosis-induction assay
Cells were stained in 1X Annexin Binding Buffer (Thermo Fisher, V13246) with Annexin-V, FITC conjugate (Thermo Fisher, A13199, 2.5 μL in 50 μL cell suspension) or Annexin-V, Pacific Blue conjugate (Thermo Scientific, A35122, 2.5 μL in 50 μL cell suspension) according to the manufacturer's protocol, and then subjected to flow cytometric analysis for quantification of apoptosis induction.

### Immunoprecipitation
For α-FLAG-immunoprecipitation, cells were washed with ice-cold PBS and then lysed with ice-cold NP-40 buffer (0.1% NP-40, 15 mM Tris-HCl pH7.4, 1 mM EDTA, 150 mM NaCl, 1 mM MgCl2, 10% Glycerol) containing protease inhibitors (Sigma, 11697498001). Cell lysates were incubated with α-FLAG gel (Sigma, A2220) for 2 h at 4 °C on a rotating wheel. α-FLAG gel was pelleted by centrifugation (1150 × *g* for 1 min at 4 °C), washed with NP-40 buffer 5 times, and then mixed with Laemmli buffer and boiled at 95 °C for 5 min.

For α-SMC1A-IPs, cells were washed with ice-cold PBS, lysed with ice-cold NP-40 buffer containing protease inhibitors, and then protein lysates were incubated with α-SMC1A antibody (Abcam, ab140493, 1–10 μg per IP) overnight at 4 °C on a rotating wheel. For each sample, 50 μL of Protein-G sepharose beads in a 50% slurry (Invitrogen, 101242) was washed 3 times with NP-40 buffer, and then incubated with the protein lysate and α-SMC1A antibody mixture for 2 h at 4 °C on a rotating wheel. Protein-G agarose beads were pelleted by

centrifugation (1150 × *g* for 1 min at 4 °C), washed with NP-40 buffer 3 times, and then mixed with Laemmli buffer and boiled at 95 °C for 5 min.

### Biotin affinity purification
Cells were washed with ice-cold PBS and then lysed with ice-cold NP-40 buffer containing protease inhibitors. Cell lysates were incubated with Pierce streptavidin-coated magnetic beads (Thermo Fisher, 88816) for 2 h at 4 °C on a rotating wheel. Streptavidin-coated magnetic beads were washed 5 times with NP-40 buffer, and then mixed with Laemmli buffer and boiled at 95 °C for 5 min.

### Cellular fractionation
Cells were washed with ice-cold PBS and first lysed with NP-40 buffer containing protease inhibitors. Supernatants were saved as cytoplasmic/nucleoplasmic soluble fraction. Remaining insoluble pellet was washed once with NP-40 buffer and then lysed with 1% SDS buffer, sonicated, and boiled at 95 °C for 10 min to extract chromatin-bound fraction.

### Western blotting
Protein samples were loaded for SDS-PAGE electrophoresis and transfer to PVDF membranes (Millipore, IPVH00010). Membranes were blocked in 5% blotting-grade blocker (Bio-Rad, 1706404) with PBST for 30 min, and then incubated with primary antibodies in 5% blocker/PBST for 2 h at room temperature on the benchtop. Membranes were washed 3 times with PBST for 10 min each and then incubated with HRP-linked secondary antibodies overnight at 4 °C on a rocker. Membranes were washed 3 times with PBST for 15 min each, 1 time with PBS for 15 min, and then visualized on films in a dark-room after the addition of ECL reagents (Thermo Fisher Scientific, Super-Signal West Pico PLUS, 34580). Primary and secondary antibodies used for Western blotting are listed in Supplementary Table 1.

### Mass spectrometry of proteomics
For the biotin affinity purified samples, $1 \times 10^8$ MV4-11 cells expressing TurboID-HA-DCAF15 were treated with 5 μM MLN4924 (MedChem Express, HY-70062) or DMSO for 6 h, and 50 μM biotin (Sigma, B4639) for 2 h. Cells were washed with ice-cold PBS and then lysed with ice-cold NP-40 buffer containing protease inhibitors. Protein lysates were incubated with 100 μL of streptavidin-coated magnetic beads for 2 h at 4 °C on a rotating wheel. Beads were washed 4 times with NP-40 buffer and then once with 50 mM Tris pH7.5 prior to freezing at −80 °C until processing for mass spectrometry. Samples were reduced, alkylated with iodoacetamide, and digested with trypsin. Tryptic digest was C18 cleaned up before LC−MS/MS analysis. For the α-FLAG-immunoprecipitated samples, in-solution digestion followed by LC−MS/MS was performed.

LC−MS/MS of tryptic peptides was performed using a nanoAC-QUITY UPLC (Waters) coupled with a Q Exactive Plus mass spectrometer (Thermo Fisher Scientific). Samples were loaded onto a UPLC Symmetry trap column (180 μm i.d. × 2 cm packed with 5 μm C18 resin; Waters), and peptides were separated by reversed-phase HPLC on a

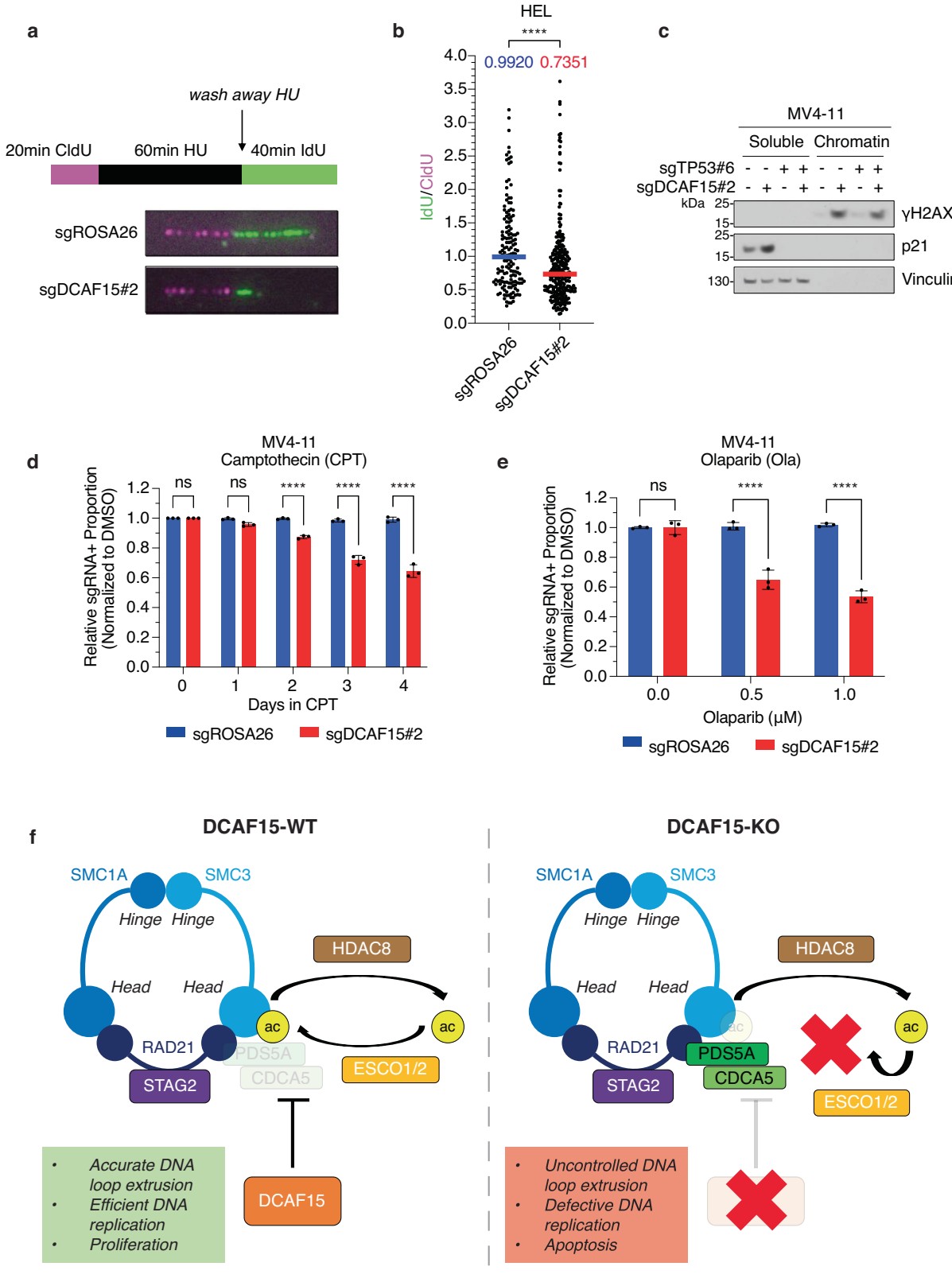

BEH C18 nanocapillary analytical column (75 µm i.d. × 25 cm, 1.7 µm particle size; Waters) using a 90 min gradient formed by solvent A (0.1% formic acid in water) and solvent B (0.1% formic acid in acetonitrile) as follows: 5–30% B over 70 min, 30–80% B over 10 min, and constant 80% B for 10 min. Eluted peptides were analyzed by the mass spectrometer set to repetitively scan m/z from 400 to 2000 in positive ion mode. Full MS spectra were recorded at a resolution of 70,000 in profile mode. Full MS automatic gain control target and maximum injection time were set to 3e6 and 50 ms, respectively. MS2 spectra were recorded at 17,500 resolution and MS2 automatic gain control target and maximum injection time were set to 5e4 and 50 ms, respectively. Data-dependent analysis was performed on the 20 most

**Fig. 7 | DCAF15 loss disrupts DNA replication fork integrity. a** Workflow of DNA fiber analysis for quantification of replication fork restart efficiency after 1 mM hydroxyurea (HU)-induced replication fork stalling. Cells cultured in the presence of CldU for 20 min prior to addition of HU. After 60 min, cells washed with PBS prior to adding Idu for 40 min. Representative images of DNA fibers shown for Cas9+ HEL cells infected with *ROSA26-* or *DCAF15*-targeting sgRNAs. **b** Quantification of (**a**) shows IdU/CldU ratios (replication fork restart efficiency) in individual experimental conditions (sgROSA26: $n = 154$ fibers analyzed, sgDCAF15#2: $n = 278$ fibers analyzed). Median IdU/CldU ratios labeled and indicated by horizontal blue (sgROSA26) and red (sgDCAF15#2) bars. Two-tailed Mann–Whitney test, ****$p$-value < 0.0001. **c** Cas9 + MV4-11 *TP53*-WT and *TP53*-knockout (sgTP53#6) cells infected with *ROSA26-* or *DCAF15*-targeting sgRNAs harvested 4 days post lentiviral sgRNA infection. Cytoplasmic (Soluble) and nuclear (Chromatin) protein fractions analyzed by Western blot for indicated proteins. Immunoblots representative of three independent experiments. **d** Competition-based proliferation assay performed in Cas9 + MV4-11 cells infected with *ROSA26-* or *DCAF15*-targeting sgRNAs were grown in presence of 1 nM camptothecin (CPT). sgRNA+ populations monitored over time for mCherry expression by flow cytometry. Plotted values are relative sgRNA+ proportions growing with CPT normalized to matched sgRNA+ growing with DMSO. Error bars mean ± SD, $n = 3$ biologically independent replicates. Two-way ANOVA (with Bonferroni's multiple comparisons test), ****padj < 0.0001, ns not-significant. **e** Same as in (**d**), except cells grown in the presence of olaparib (Ola) for 7 days. Relative sgRNA+ proportion quantification is shown for cells growing with olaparib normalized to matched sgRNA+ proportion growing with DMSO. Error bars mean ± SD, $n = 3$ biologically independent replicates. Two-way ANOVA (with Bonferroni's multiple comparisons test), ****padj < 0.0001, ns not-significant. **f** HDAC8-mediated cohesin deacetylation enables CRL4-DCAF15 recruitment. DCAF15-mediated destabilization of cohesin-bound PDS5A and CDCA5 enables timely ESCO1/2-mediated cohesin re-acetylation, vital for accurate halting of DNA loop extrusion, efficient DNA replication fork progression, and sustained AML proliferation (Left). In the absence of DCAF15, HDAC8-de-acetylated cohesin is inappropriately hyper-loaded with PDS5A and CDCA5, which precludes ESCO1/2-mediated cohesin re-acetylation and leads to uncontrolled DNA loop extrusion, defective DNA replication fork progression, and activation of apoptosis (Right). Source data are provided as a Source Data file.

abundant ions using an isolation width of 1.5 m/z and a minimum threshold of 1e4. Peptide match was set to preferred, and unassigned and singly charged ions were rejected. Dynamic exclusion was set to 30 s.

Peptide sequences were identified using MaxQuant (v1.6.17.0)[77]. MS/MS spectra were searched against a Swiss Prot human protein database (6/4/2021) and a common contaminants database. Precursor mass tolerance was set to 4.5 ppm in the main search, and fragment mass tolerance was set to 20 ppm. Digestion enzyme specificity was set to Trypsin/P with a maximum of 2 missed cleavages. A minimum peptide length of 7 residues was required for identification. Up to 5 modifications per peptide were allowed; acetylation (protein N-terminal), deamidation (Asn), and oxidation (Met) were set as variable modifications, and carbamidomethyl (Cys) was set as a fixed modification. Peptide and protein false discovery rates (FDR) were both set to 1% based on a target-decoy reverse database.

For the biotin affinity purification LC–MS experiment, 2 total samples were analyzed ($n = 1$ replicate each), with the DMSO-treated sample serving as a baseline control for comparison with the MLN4924-treated sample. For the α-FLAG-immunoprecipitation experiment, 2 total samples were analyzed ($n = 1$ replicate each), with the FLAG EV sample serving as a non-specific background control for comparison with the FLAG-DCAF15 sample. Mass spec data is available in Supplementary Data 3.

### Cycloheximide-chase assay
$1 \times 10^7$ control or *DCAF15*-knockout HEL cells were plated in 15-cm plates and treated with 5 µg/mL cycloheximide (Millipore Sigma, C7698) for 0, 0.5, 1, 2, 4, and 8 h, respectively, prior to α-SMC1A-IPs.

### MG132, MLN4924, and indisulam treatments
Prior to collecting cells, cells were treated with: 10 µM MG132 (Peptides International, IZL-3175-v) for 6 h; 5 µM MLN4924 (MedChem Express, HY-70062) for 6 h; 3 µM indisulam (Millipore Sigma, SML1225-5MG) for 6 h.

### Cell cycle synchronization
Cell cycle synchronization at the G1/S checkpoint was achieved by double thymidine block, wherein cells were cultured in the presence of 2 mM thymidine (Sigma–Aldrich, T1895-5G) for 16 h, washed 2× with PBS and released in fresh media for 8 h, then subjected to a second round of thymidine incubation (2 mM; 16 h).

Unless otherwise indicated in Figure Legend, to synchronize cells in the G2 phase, cells pre-synchronized at G1/S by double thymidine block were washed 2× with PBS and released in fresh media for 3 h,

prior to culturing in the presence of 5 µM Ro-3306 (Selleck Chemicals, S7747) for 9 h.

To synchronize cells in prometaphase (PM), cells pre-synchronized at G1/S by double thymidine block were washed 2× with PBS and released in fresh media for 3 h, prior to culturing in the presence of 150 nM nocodazole (Selleck Chemicals, S2775) for 9 h.

### In vitro ubiquitination assay
In vitro ubiquitination assay was performed in a volume of 150 µL, containing 50 mM Tris pH7.6, 5 mM MgCl$_2$, 2 mM ATP (Roche, 10127523001), 1.5 ng/µL UBE1 (Boston Biochem, E-304-050), 10 ng/µL UBE2G1 (Boston Biochem, E2-700-100), 10 ng/µL UBE2D3 (Boston Biochem, E2-627-100), and 2.5 µg/µL ubiquitin (Boston Biochem, U-100H-10M), which was added to 90 µl of PBS-washed bead-bound FLAG-immunoprecipitated FLAG-DCAF15 bound to HA-SMC1A, HA-PDS5A, HA-CDCA5, and other endogenous cohesin complex proteins from HEK293T cells. The reactions were incubated at 30 °C, and 60 µL of the total 240 µL reaction volume was collected after 0, 30, and 60 min of incubation. Samples were mixed with Laemmli buffer, boiled at 95 °C for 10 min, subjected to SDS-PAGE, and analyzed by immunoblot.

### DNA fiber assay
~0.1–0.3 × 10$^6$ asynchronously growing HEL cells were sequentially labeled with 20 µM CldU (Millipore Sigma, C6891) and 200 µM IdU (Millipore Sigma, I7125) thymidine analogs. To assess replication fork restart, control and *DCAF15*-knockout HEL cells were labeled with CldU for 20 min followed by incubation in hydroxyurea (1 mM) for 60 min (Millipore Sigma, H8627). Cells were washed three times with warm 1X PBS and released in IdU-containing media for another 40 min. Cells were trypsinized and washed two times with ice-cold 1× PBS, and ~2000 cells were spotted and lysed (200 mM Tris-HCl pH7.4, 50 mM EDTA, 0.5% SDS) on silane-coated slides (Newcomer Supply, 5070). DNA fibers were stretched along the slide by gravity, prior to air-drying and fixation (3:1 methanol:acetic acid). Fibers were denatured with 2.5 M HCl for 1 h at room temperature. Slides were neutralized with 400 mM Tris-HCl pH7.4, washed with PBST, and blocked in 5% BSA and 10% goat serum overnight at 4 °C. Subsequently, slides were incubated with rat α-CldU antibody (Abcam, ab6326, 1:200) for 1 h at room temperature (RT). After 3 washes with PBST, slides were incubated with AlexaFluor 647-conjugated α-rat IgG secondary antibody (Thermo Fisher, A-21247, 1:100) for 1 h at RT followed by stringent PBST washing. Similarly, slides were further incubated with mouse α-IdU antibody (BD Pharmigen, 347580, 1:40) and AlexaFluor 488-conjugated α-mouse IgG secondary antibody (Thermo Fisher, A-11001, 1:100) for 1 h each at RT. Finally, slides were mounted with Prolong

Gold antifade mountant (Thermo Fisher, P36930). Labeled DNA fibers were examined using Nikon Eclipse 80i fluorescence microscope with a 60X Oil DIC N2 objective lens. Cy5 and GFP filters were used to detect CldU- and IdU-labeled fibers, respectively. Fibers were analyzed and scored using Fiji software. To negate the potential effect of fork progression defects caused by differing experimental conditions on restart efficiency quantification, the length of an IdU tract was normalized to that of the matched CldU tract for a given replication fork. At least 150 untangled fibers were scored for each experimental condition. The Mann–Whitney test was used to determine statistical significance.

### RNA-seq

Total RNA was extracted from $2 \times 10^6$ cells using the RNeasy Mini Kit (QIAGEN, 74104). 2 μg total RNA per sample was shipped to GENEWIZ (Azenta Life Sciences) for Standard RNA-Seq service. Briefly, polyA+ transcripts were isolated, RNA-seq library was prepared, and 3 biological replicates per experimental condition were sequenced on an Illumina NovaSeq or HiSeq machine. Reads were mapped and analyzed with a bioinformatic pipeline based on STAR RNA-seq Aligner, SAMtools, and DESeq2. Human genome version GRCh38 was used. RNA-seq data is available in Supplementary Data 2.

### ChIP-seq

$2 \times 10^7$ cells were collected, washed, and cross-linked with 1% paraformaldehyde in PBS for 10 min and quenched with 125 mM glycine for 5 min at room temperature. Fixed cells were suspended in lysis buffer (0.1% SDS, 1% Triton X-100, 10 mM Tris-HCl, 1 mM EDTA, 0.1% NaDOC, 300 mM NaCl, 0.25% sarkosyl, 1 mM DTT, and protease inhibitors) and sonicated in Covaris Ultrasonicator. Samples were pre-cleared with 10 μL of BSA-coated Protein A Dynabeads (Thermo Fisher, 10001D) at 4 °C for 1 h and incubated with 20 μg of antibody overnight. The following antibodies for ChIP-seq were used: α-Total-SMC3 (Bethyl Laboratories, A300-060A, 20 μg per IP) and α-acetyl SMC3 (Lys105/106) (EMD Millipore, MABE1073, 20 μg per IP). The antibody-conjugated samples were incubated with 75 μL of BSA-coated Protein A Dynabeads at 4 °C for 1 h. The immunoprecipitants were washed twice with low-salt buffer (150 mM NaCl, 0.1% SDS, 1% Triton X-100, 1 mM EDTA, and 50 mM Tris-HCl), twice with high-salt buffer (500 mM NaCl, 0.1% SDS, 1% Triton X-100, 1 mM EDTA, and 50 mM Tris-HCl), twice with LiCl buffer (150 mM LiCl, 0.5% Na-Deocycholate, 0.1% SDS, 1% NP-40, 1 mM EDTA, and 50 mM Tris-HCl), and once with TE buffer (1 mM EDTA, 10 mM Tris-HCl). Bound DNA was eluted with 200 μL of elution buffer (1% SDS, 200 mM NaCl, 10 mM EDTA, and 50 mM Tris-HCl) at 65 °C overnight. Eluted DNA was incubated with 2 μL of 0.5 mg/mL RNase A (Fisher Scientific, BP2539250) at 37 °C for 1 h, incubated with 2 μL of 20 mg/mL Proteinase K (Thermo Fisher, EO0491) at 55 °C for 1 h, and purified with QIAquick PCR purification kit (Qiagen, 28104).

For ChIP-seq, ChIP-seq libraries were prepared with NEBNext Ultra II DNA library prep with sample purification beads (NEB, E7103S) following the manufacturer's instructions. The libraries were sequenced using Illumina NextSeq 2000. FASTQ files were aligned with Bowtie2, reads were sorted and duplicates removed with SAMtools. Peaks were called with MACS2[78]. Further analysis was performed in RStudio using the R packages Rsamtools, GenomicAlignments, soGGi, ggplot2, and profileplyr. Human genome version h19 was used. ChIP-seq data is available in Supplementary Data 4.

### Hi-C

Hi-C was performed on $1 \times 10^6$ cells per replicate using the Arima High Coverage Hi-C Kit (Arima Genomics, A101030), following all manufacturer instructions. Hi-C libraries were prepared using the NEBNext Ultra II DNA Library Prep Kit (New England BioLabs, E7645S) and NEBNext Dual Index Primers (New England BioLabs, E7600S), with modifications to the protocol as specified by the Arima High Coverage Hi-C Kit. Libraries were sequenced on an Illumina NextSeq 2000 instrument using $55 \times 2$ paired-end cycle settings.

### Hi-C read processing and analysis

Hi-C reads were combined between replicates and aligned to the human hg19 genome with Juicer[79] using an annotation file based on the Arima restriction enzymes and default settings. Contact matrices were generated by Juicer in.hic format using Knight-Ruiz (KR) normalization[80] and converted to.cool format at various resolutions using the hicConvertFormat function in HiCExplorer[81]. Format conversion was performed over two sequential hicConvertFormat calls, first converting.hic to.cool, then converting.cool to.cool using the --correction_name KR option to retain KR-normalized values. KR-normalized data was used for all downstream analysis. Contact probability plots were generated using hicPlotDistVsCounts (HiCExplorer). Hi-C heatmaps were generated using hicPlotMatrix and hicPlotTADs (HiCExplorer). To identify contact boundaries and domains,.cool files were converted to.h5 with hicConvertFormat, then hicFindTADs (HiCExplorer) was run on 50-kb resolution data with options --minDepth 150000 --maxDepth 1000000 --step 50000 --thresholdComparisons 0.01 --correctForMultipleTesting fdr. Chromatin loops were identified using HiCCUPS CPU (Juicer). sgDCAF15#2 loops were designated as shared loops if the left anchor overlapped the left anchor of a loop called in sgROSA26 and the right anchor overlapped the right anchor of the same sgROSA26 loop, and as gained loops otherwise. The full set of loops was defined as sgROSA26 loops plus loops gained in sgDCAF15#2. Pileup plots of contact domains and loops were generated using coolpup.py[82] with 10-kb resolution data and settings --rescale --local --ignore_diags 2 for contact domains and --flank 100000 --ignore_diags 0 --mindist 0 for loops.

### Statistical analyses

Statistical analysis was performed using either Graphpad Prism or R. All graphs display mean values with standard deviation (SD) error bars, as indicated in respective Figure Legends. Reproducibility and sample size numbers for each Figure are listed in Figure Legends. Two-tailed unpaired Student's t-test, two-tailed Mann–Whitney test, Fisher's exact test, or one-way or two-way ANOVA analyses were performed, as indicated in respective Figure Legends.

### Reporting summary

Further information on research design is available in the Nature Portfolio Reporting Summary linked to this article.

## Data availability

Next-generation sequencing data are deposited in the Gene Expression Omnibus (GEO) database with the accession number GSE241581. Mass spectrometry proteomics data have been deposited in the MassIVE data repository under ID MSV000092697 [https://massive.ucsd.edu/ProteoSAFe/dataset.jsp?task=fde90091ea2c4f4c86f69299de2f830e] and have been deposited at the ProteomeXchange Consortium with the dataset identifier PXD044661. Source data are provided with this paper.

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

## Acknowledgements

We thank Amelia Nguyen, Jennefer Hernandez, Akash Sinha, and Khai M. Nguyen for technical experimental assistance. We thank Sehbanul Islam for assistance with qRT-PCR quantification of Hi-C libraries. We thank Reyaz ur Rasool and Erick Mitchell-Velasquez for assistance with Illumina sequencing. We thank Qinglan Li for bioinformatics assistance. We thank Gerald B. Wertheim and Martin P. Carroll for expert experimental advice. We thank Susannah Rankin for kindly gifting the α-ESCO1 antibody. We thank Aaron D. Viny and Zuzana Tothova for fruitful conversations regarding cohesin complex biology and AML disease relevance. We thank the Penn Cytomics and Cell Sorting Resource Laboratory and the Children's Hospital of Philadelphia (CHOP) Flow Cytometry Core for help with fluorescence-activated cell sorting (FACS). We thank Dorothy Hunter, Jason Cutrera, and the Research Facility and Lab Services staff for experimental equipment support. We thank Ashley N. Hughes and Lesley Moreno for administrative support. K.S. acknowledges support from R01GM143229. J.S. acknowledges support from NIH/NCI (R01CA258904). This work was supported by grants from the American Cancer Society (RSG-19-199-01), National Cancer Institute (2R01CA207513), and National Heart, Lung, and Blood Institute (R01HL159175) to L.B. L.B. is a scholar of the Leukemia & Lymphoma Society.

## Author contributions

G.P.G. conceived of, designed, and performed most of the experiments, and wrote the manuscript. R.C. generated dTAG-DCAF15 cell lines, performed DCAF15-SMC1A interaction mapping experiments, dTAG-DCAF15 RNA-seq experiment, ESCO1 ChIP-qPCR experiment, and γH2AX immunofluorescence experiments. Z.C. performed CRISPR library construction, and screening of MOLM-13, MV4-11, Jurkat, OPM-1, and Hep-G2 cell lines. N. Zhou performed CRISPR screening of U-2932 cell line and assisted with ChIP-seq experiment. M.M. validated dTAG-DCAF15 cell lines and performed DCAF15-SMC1A interaction validation experiments. A.D. performed DNA fiber assay experiment. P.W. assisted with Hi-C experiment and performed analysis of Hi-C data. T.B. performed mass spec analyses of FLAG-DCAF15 and TurboID-DCAF15 interactomes. B.W. performed computational prediction modeling of the DCAF15-SMC1A interaction. N. Zheng oversaw computational prediction modeling of the DCAF15-SMC1A interaction. H-Y.T. oversaw mass spec analyses of FLAG-DCAF15 and TurboID-DCAF15 interactomes. K.S.

oversaw Hi-C experiment and analysis of Hi-C data. R.A.G. oversaw the DNA fiber assay experiment. J.S. oversaw CRISPR library construction and screening and provided LRG2.1 and LRCherry2.1 vectors and spCas9+ human AML cell lines. L.B. conceived of and directed the project, oversaw the results, revised the manuscript, and secured the funding.

## Competing interests

R.A.G. is a co-founder of RADD Pharmaceuticals and an advisor to Dong-A ST and TrevarX Biomedical, Inc. N.Zheng is a co-founder of and has financial interests in Seed Therapeutics and Molecular Glue Labs Ltd. N. Zheng also serves as a member of the scientific advisory board of SyntheX with financial interests. The remaining authors declare no competing interests.
