## [Peer Review File · Nature Communications]

DCAF15 control of cohesin dynamics sustains acute myeloid leukemiaREVIEWER COMMENTS

Reviewer #1 (Remarks to the Author):

In this study, Grothusen et al. identified DCAF15 as an important regulator of AML proliferation by mediating ubiquitination and degradation of two regulatory subunits of the cohesin complex, PDS5A and CDCA5. They demonstrate that PDS5A and CDCA5 protein degradation interferes with proper cohesin complex acetylation and cohesin-DNA interaction.

In brief, the authors performed a domain-focused CRISPR/Cas9 dropout screen targeting the domain mediating the interaction between the Cullin-RING ligase receptor and the Cullin module across a panel of cancer cell lines, and compared 2 MLL-rearranged AML cell lines versus 4 non-AML cell lines (ALL, DLBCL, MM and HCC). They focused on one of the AML-specific hits – DCAF15, whose high expression was also associated with poor OS in AML, and whose endogenous targets were poorly understood, but which has a lot of therapeutic promise (interaction with the anti-cancer aryl-sulfonamide molecular glue compounds). The authors showed that DCAF15 has a p53-dependent, pro-proliferative function in AML, by protecting replication fork integrity. Using a number of different biochemical approaches, the authors showed that this function is mediated by interacting with the cohesin complex and regulating its acetylation status and therefore its recycling dynamics.

Overall, the authors report a novel and interesting mechanism of regulation of cohesin dynamics, which acts at the post-translational level to modulate the abundance of cohesin regulatory factors (like PDS5A and CDCA5) and therefore fine tune cohesin recycling. The biochemistry and chromatin experiments are well done, well controlled and solidly support the main authors' claims. The cell biology part that supports the physiological function of DCAF15 is less well developed and the DNA damage data are not sufficient to draw a direct conclusion between replication fork restart and DNA damage accumulation. The paper is well written and the figures are nicely laid out. I have summarized my comments below:

Major points:

1. Figure 1H – this experiment is missing a control (sgROSA26), and extrapolating from Figure 1F, expression of DCAF15(ntG207C) seems to lead to only a partial rescue. How do the authors explain this result? Have they checked the level of expression of the mutant version of DCAF15? Perhaps too high of an expression of DCAF15 may be too toxic or the mutant version of DCAF15 is not equivalent of the wild type DCAF15? This partial rescue was similarly seen in panel 1G with regards to RBM39 degradation.
2. Figure 2A – I was somewhat surprised by the very low number of differentially expressed genes upon DCAF15 loss. Have the authors examined the effects in a shorter-term experiment (eg. with degron approaches)? It would be helpful to highlight key modulated genes in this plot, especially since there appear to be so few.
3. Figure 2B - Are the p values in Figure 1B adjusted p values/FDR?
4. Figure 2D - The cleaved Caspase3 blots are not very convincing. A couple of other assays addressing apoptosis and cell death should be performed.
5. Figure 2F – Can the authors elaborate more on the reasons behind the partial rescue with sgRNA targeting TP53 in this competition assay? The authors may want to consider performing cell cycle analysis in the context of sgTP53 rescue experiments as well. Indeed, the very slight increase in Annexin V+ cells I panel 2G would unlikely explain the drastic loss of DCAF15 KO cells.
6. The biochemical approaches used by authors in Figure 3 are very nice, in particular IP-MS and proteomics in the presence and absence of MLN4924. Can the authors comment on why they decided to do the FLAG-IP in 293T cells instead of the more relevant MV4-11 cells used throughout the paper? Did the authors encounter toxicity in the MV4-11 cells?
7. Is there a way to establish whether there is a competition between DCAF15 and RAD21 that could destabilize the cohesin ring?
8. DCAF15 KO cells are definitively sensitive to DNA damaging agents. The higher sensitivity to the

PARP inhibitor olaparib indicates a putative deficiency in Homologous Recombination (HR), as HR-deficient cells rely on PARP to repair DNA damage. Furthermore, DNA damage occurring at replication forks is mainly repaired via HR. Since cohesins have now been involved in DNA damage responses to favor the use of sister chromatids, and stall replication in order to prevent amplification of damaged DNA, it would be interesting to further investigate that aspect.

Minor points:

1. I would suggest better coordination of color coding throughout the manuscript to ease interpretation of various figures.
2. The use of an endogenously tagged DCAF15 would strengthen a lot of the biochemistry experiments described.
3. The introduction could be significantly shortened.

Reviewer #2 (Remarks to the Author):

The authors demonstrated a novel mechanism of DCAF15 required for proliferation of acute myeloid leukemia (AML). The main findings from this manuscript are: 1) DCAF15 is required for DNA damage reduction via efficient replication fork integrity, resulting in improvement of sensitivity to replication stress-inducing therapeutics in DCAF15-deficient AML. 2) DCAF15-containing CRL4 ubiquitin E3 ligase complex interacts with SMC1A in cohesion complex and targets PDS5A and CDCA5 which are known as the cohesion regulatory factors. These findings are novel and significant and suggest novel functions of DCAF15 in sustaining AML proliferation via post-translational control of cohesion dynamics. The experiments are logical, clearly explained and high quality. The conclusions drawn from the results are justified and quite thought provoking. I recommend publication with minor questions.

Minor point: The authors checked DCAF15 knockout by detecting RBM39 degradation due to no effective commercial antibody against the DCAF15. However, published journals (for example, PMID: 33833131) clearly show a DCAF15 protein levels using DCAF15 antibodies (cat no. SAB1103260, Sigma). It would be great if the author can detect DCAF15 directly using above antibodies. If the antibody is available, it would be nice if the protein level or subcellular localization of DCAF15 can be changed during cell cycle.

Reviewer #3 (Remarks to the Author):

The authors identify DCAF15 as an AML-109 biased CRL dependent gene by domain-focused CRISPR/Cas9-based knockout (KO) selection. By RNA-seq and double KO, they find DCAF15 loss suppresses AML via activation of p53 tumor suppressive programs and induction of apoptosis. Next, they perform IP-MS and TurboID and identify SMC1A and SMC3 as interactors of DCAF15 E3, which degrades their neighbors PDS5A and CDCA5 to facilitate re-acetylation of cohesin in the next cycle. The overall manuscript presents some interesting results although the main mechanistic insight needs further solid evidence.

Major concerns:

Fig. 3G and H, AlphaFold and IP results show that DCAF15 interacts SMC1A C-terminus. Is this interaction direct? For instance, confirm it by pulldown assays.

Fig. S3D and E, Both DCAF15 and RAD21 use similar alpha-helix to bind at the same site on SMC1A C-

terminal domain. If so, does DCAF15 compete with RAD21 for SMC1A binding and interfere with cohesin complex formation?

The above two points raise another concern together with IP results shown in Fig. 4A, Fig. S4C, depletion of SMC3 or RAD21 decreased interaction of SMC1A with DCAF15, i.e., DCAF15 accesses cohesin proteins assembled on the intact ring-shape structure. At least, the authors should explain how DCAF15 accesses SMC1A head domain which is directly bound by RAD21 in the cohesin complex. Related to this, in Fig. 5A, SMC3ac blocks DCAF15 binding to SMC1A, HDAC8-mediated SMC3-deacetylation, occurring in G2/M phases, precedes DCAF15 binding to the cohesin complex. HDAC8 de-acetylates cohesin which then released from chromosome in anaphase. RAD21 is cleaved by separase in anaphase. The author proposed that DCAF15 interact with SMC1 relies on the full cohesin complex. All these pieces should be put together.

DCAF15-cohesin interaction mutants should be phenotyped to validate the cellular defects observed in DCAF15 KD are due to interaction with cohesion.

Fig. S4A and B, as well as Fig. S3A, the authors exclude SMC1 and p53 as the substrates of DCAF15 E3 by DCAF15 KD. Controls? An alternative way is to test the effects by overexpression DCAF15. Different substrates may require different threshold levels of E3. Because these are critical mechanistic points, we should be very cautious to exclude them by a single experiment.

Fig4A and B, are total levels of PDS5A and CDCA5 changed upon DCAF15 KD? PDS5 protein level during cell cycle should be checked, it has been shown stable during cell cycle (Naif Al-Jomah, 2020).

In Figure 4C and E, only PDS5A, but not CDCA5, is detected with poly-Ubi, indicating that CDCA5 instability may be an indirect downstream effect of PDS5A degradation. CDCA5 degradation is known to be mediated by multiple sites phosphorylation then by the APC complex. Moreover, did the authors test WAPL, which is also mediated by PDS5A?

Fig. 5B, the authors conclude that DCAF15 binds to cohesin that has been de-acetylated by HDAC8 and catalyzes ubiquitylation/degradation of PDS5A and subsequent release of CDC5A; thus, refreshing cohesin for ESCO1/2-mediated re-acetylation upon reloading onto chromatin. This conclusion is important and needs further evidence, for instance, using an inducible degron system like AID2, to deplete DCAF15 in G2/M and examine the SMC3 and SMC3ac levels in the next cycle. This system is also important for experiments shown in Fig7. PDS5A and CSC5A are recruited to cohesin in a SMC3ac-dependent way, does deacetylation also contribute to their dissociation and DCAF enhances this process?

Fig. 5D-G, did the authors examine ESCO2/1 simultaneously? This will tell us how DCAF15 affects SMC3ac, e.g., DCAF15 blocks their recruitment?

Fig7C, Is ATM-CHK2 activated? Can ATM and CHK2 inhibitors rescue DCAF15 KO cells like p53 inactivation?

Does DCAF15 functions in sister chromatid cohesion or separation?

Minor issues:

Figure S1A does not clearly show which dot is DCAF15.

Reviewer #4 (Remarks to the Author):

The authors of the study present here a novel role for DCAF15 in regulating the cohesin complex. The leukemic cell line MV4-11 was used in most of the experiments. MV4-11 presents a t4;11 (MLL

rearrangement) and FLT3-ITD mutation. MLL rearrangements occur in approximately 5% of Acute myeloid leukemia and FLT3 mutations in approx. 30%.

Please see comments below:

The number of differentially regulated genes captured by the RNAseq seems very low fig2A and 2E, could the authors comment?

Fig2D WB, it is not clear how many times this experiment was performed, 3 independent experiments are expected with quantification.

Fig2G apoptosis is increased by less than 2% in DCAF15 KO, it is unlikely to explain the two fold differences observed in the proliferation assay. Cell cycle analyses would reveal whether the phenotype is mostly due to an impact on cell cycle regulation.

DCAF15 is expected to be mainly expressed in the cytoplasm and not in the nucleus where the cohesin complex is found. In order to confirm that DCAF15 plays a role on the cohesin complex, nuclear localisation of DCAF15 should be demonstrated by immunofluorescence experiments. These experiments would also allow the demonstration of the co-localisation in the nucleus of DCAF15 with different cohesin proteins.

It would be interesting to observe whether protein or RNA levels of the different components of the cohesin complex are affected by DCAF15 KO?

The link between ac-SMC3 and DCAF15-destabilises cohesin-bound PDS5A and CDCA5 should be further investigated to confirm that DCAF15 impacts ac-SMC3 via its role on PDS5A and CDCA5.

Fig7C gH2AX should be quantified by ImmunoFluorescence (scoring the Foci)

Fig7D DCAF15 KO cells already have a proliferative disadvantage (see fig1F), therefore it is not clear what is the effect of CPT. A synergistic experiment should provide more insights in the cumulative or synergistic effect of CPT and DCAF15 loss. The same question is raised by the following figure using PARP inhibitors.

Most of the work has been carried out on one cell line the MV4-11, although the authors present strong evidences of a potential role of DCAF15 on the cohesin complex, the use of one cell line is not sufficient to confirm a pan effect of DCAF15 on AML which is a highly heterogeneous disease.

In addition, to determine whether DCAF15 is a safe target, similar experiments should be carried out on normal hematopoietic stem/progenitor cells to determine tolerance and toxicity.

04/19/2024

Enclosed is the revised version of our manuscript (Grothusen and Chang *et al.*, #NCOMMS-23-35817-T), in which we have addressed reviewer comments as outlined in your initial decision letter.

Reviewer Comments:

Reviewer #1 (Remarks to the Author):

The Reviewer #1 points out that *“Overall, the authors report a novel and interesting mechanism of regulation of cohesin dynamics, which acts at the post-translational level to modulate the abundance of cohesin regulatory factors (like PDS5A and CDCA5) and therefore fine tune cohesin recycling. The biochemistry and chromatin experiments are well done, well controlled and solidly support the main authors’ claims. The cell biology part that supports the physiological function of DCAF15 is less well developed and the DNA damage data are not sufficient to draw a direct conclusion between replication fork restart and DNA damage accumulation. The paper is well written and the figures are nicely laid out. I have summarized my comments below”*. She/He points out the following concerns:

Major points:

1) *Figure 1H – this experiment is missing a control (sgROSA26), and extrapolating from Figure 1F, expression of DCAF15(ntG207C) seems to lead to only a partial rescue. How do the authors explain this result? Have they checked the level of expression of the mutant version of DCAF15? Perhaps too high of an expression of DCAF15 may be too toxic or the mutant version of DCAF15 is not equivalent of the wild type DCAF15? This partial rescue was similarly seen in panel 1G with regards to RBM39 degradation.* We appreciate the opportunity provided by the Reviewer to elucidate the details presented in **Figure 1H**. In this figure, all time points have been normalized based on the sgROSA26 control and on Day 6 post-infection, as indicated by the y-axis label. This normalization was necessary because cells expressing MIGR1(*DCAF15-ntG207C*) tend to exhibit lower infection efficiency of the sgRNA as

Figure 1. Competition-based proliferation assay performed in MV4-11 Cas9+ cell lines stably expressing empty vector (EV) (Left) or DCAF15[ntG207C] (Right) and infected with lentiviruses containing ROSA26-targeting (negative control) or DCAF15-targeting sgRNAs. sgRNA+ populations were monitored over time for mCherry expression by flow cytometry. sgRNA+ proportions normalized to day 6 post lentiviral sgRNA infection were monitored over 24 days. Two-way ANOVA (with Bonferroni's multiple comparisons test), ****= $p_{adj} < 0.0001$, **= $p_{adj} < 0.015$, ns=not-significant.

reduced cell proliferation, whereas sgDCAF15 expression in the DCAF15(ntG207C) group does not significantly impact proliferation. It's important to note that the fold change difference observed between the MIGR1(EV) and MIGR1(DCAF15ntG207C) groups is attributable to the approximately 3-fold lower initial number of infected cells in the MIGR1(DCAF15ng) group.

With regards the partial rescue observed in Figure 1G, the western blot (WB) results demonstrate that the reintroduction of the DCAF15(ntG207C) variant in cells, where DCAF15 is targeted by specific sgRNA, leads to only a partial rescue of RBM39 degradation by indisulam. Quantification of the bands indicates a reduction of 90% in the MIGR1(EV) cells as compared to a 60% reduction upon reintroduction of MIGR1(DCAF15ntG207C).

As noted by the Reviewer, this partial rescue in DCAF15 E3 ligase activity aligns with the observations of partial cell proliferation rescue shown in **Figure 1H**. To provide clarity on this point, we have incorporated a statement in the Results section emphasizing that the observed partial rescue could be attributable to the restoration of partial ligase activity of DCAF15: "Validating the on-target effect of DCAF15-targeting sgRNAs, a CRISPR-resistant DCAF15 cDNA ectopically expressed (Fig. S1G) in MV4-11 DCAF15^{-/-} cells was capable of partially rescuing indisulam-mediated RBM39 degradation (Fig. 1G) and the proliferation defect upon DCAF15 loss (Fig. 1H), likely attributable to restoration of partial ligase activity of DCAF15."

2) Figure 2A – I was somewhat surprised by the very low number of differentially expressed genes upon DCAF15 loss. Have the authors examined the effects in a shorter-term experiment (e.g. with degron approaches)? It would be helpful to highlight key modulated genes in this plot, especially since there appear to be so few.

The RNA sequencing experiment depicted in Figure 2A was deliberately scheduled for 4 days post-infection. This timing ensures that Cas9-mediated gene editing at the DCAF15 locus does not affect cell proliferation (see Figure 1F and Figure 2H). Our objective was to observe the initial transcriptional response before cell proliferation and apoptosis mechanisms become predominant. Following the Reviewer's suggestion, we engineered OCI-AML3 cells to express an N-Terminal FLAG-HA-FKBP12(F36V) fusion protein at the DCAF15 locus

compared to MIGR1(EV) cells. Consequently, in a mixed culture of infected and non-infected cells, the predominantly non-infected cells can outcompete the infected ones over time. To offer a different perspective on the data, we have plotted the results by segregating the groups based on overexpression (EV vs. DCAF15 ntG207C)), as shown in **Figure 1** of this letter. Aligning with our prior findings, sgDCAF15 in the EV group leads to

(**Figure S2A**). Treatment with dTAG-V1 in OCI-AML3-FKBP12(F36V)-DCAF15 cells led to a rapid decrease in DCAF15 levels. Using these cells, we conducted RNA sequencing at 24 and 72 hours post dTAG-V1 treatment. The results, illustrated in **Figure S2C**, consistently show a significant upregulation of the p53 pathway as early as 24 hours, with greater significance at 72 hours (**Figure S2C**), aligning with data from our previous CRISPR-mediated *DCAF15* knockout experiments (**Figure 2A**). Collectively, these findings suggest that p53 activation is an immediate response to DCAF15 depletion.

3) **Figure 2B - Are the p values in Figure 1B adjusted p values/FDR?** We have remade the graph now showing the adjusted p-values (**Figure 2B**).

4) **Figure 2D - The cleaved Caspase3 blots are not very convincing. A couple of other assays addressing apoptosis and cell death should be performed.** We have measured apoptosis as Annexin-V positive cells and shown that loss of DCAF15 increases Annexin-V staining in *TP53*-competent cells (**Figure 2G** and **2I**). We extended this investigation to include the effect of DCAF15 loss on apoptosis induced by camptothecin (CTP) (**Figure S7E**). The new data suggest again that apoptosis is increased in *DCAF15*-KO cells, and the increase is exacerbated upon CPT treatment in a p53-dependent manner. In addition to apoptosis induction, as detailed in point #5, we have further assessed the role of DCAF15 in cell proliferation (please, refer to response in point #5).

5) **Figure 2F – Can the authors elaborate more on the reasons behind the partial rescue with sgRNA targeting TP53 in this competition assay? The authors may want to consider performing cell cycle analysis in the context of sgTP53 rescue experiments as well. Indeed, the very slight increase in Annexin V+ cells I panel 2G would unlikely explain the drastic loss of DCAF15 KO cells.**

We repeated the experiment by using additional *TP53* and *DCAF15* sgRNAs and observed similar rescues as presented in the original submission (**Figure 2F**). It's possible that DCAF15 has additional functions separable from the one we show here that is dependent upon p53. We have added a comment that the partial rescue might be related to a p53-independent function of DCAF15. We have commented in the text: "*AML cells transduced with two different sgRNAs targeting TP53 largely rescued the proliferation defect induced by DCAF15 loss mediated by transduction with two different sgRNAs targeting DCAF15 (Fig. 2F). Lack of complete proliferation rescue may suggest that DCAF15 possesses p53-independent pro-proliferative functions as well.*"

We performed the cell cycle analysis as suggested by the Reviewer, however no major changes in cell cycle distribution were observed upon DCAF15 loss with or without p53 loss. As such, to more directly examine the proliferation deficit caused by DCAF15 loss, we sorted MV4-11 and OCI-AML3 cells infected with either sgRosa26 or sgDCAF15 and conducted a non-competitive proliferation assay (**Figure S1H**). In this assay, cells were counted and replated at the original density every 3-4 days, offering a more direct measure of proliferation impairment. Through this additional method, we confirmed that the loss of DCAF15 results in a significant reduction in cell proliferation rate over time. Overall, the deficit of DCAF15 KO cells is a combination of both slower cell proliferation as well as higher apoptosis. We have commented in the text: "*Altogether, these results demonstrated that DCAF15*

loss suppresses AML via activation of p53 tumor suppressive programs, and that the growth defect of AML cells lacking DCAF15 may be attributable to a combination of both reduced cell proliferation rate as well as induction of apoptosis."

Finally, to address the Reviewer's feedback on the "very slight increase in Annexin V+ cells I panel 2G would unlikely explain the drastic loss of DCAF15 KO cells", we would like to reiterate that **Figure 2F** (as well as **Figure 1F**) illustrates a competition assay. In this experiment, infected cells (mCherry-positive) are mixed with non-infected cells (mCherry-negative), and the ratio of these cell populations is monitored over time. It's important to note that every 3 days, cells are split at a 1:10 ratio, which inherently favors cells with higher proliferation potential (mCherry-negative). Consequently, after several passages, the ratio of mCherry-positive to mCherry-negative cells decreases. Over a span of 25 days, this assay demonstrates that non-infected cells can outcompete cells infected with sgDCAF15. As such, the interpretation of this experiment is that DCAF15-KO cells can be outcompeted by DCAF15-competent cells in the timeframe analyzed, consistent with slower proliferation rate and increased apoptosis as explained above.

6) The biochemical approaches used by authors in Figure 3 are very nice, in particular IP-MS and proteomics in the presence and absence of MLN4924. Can the authors comment on why they decided to do the FLAG-IP in 293T cells instead of the more relevant MV4-11 cells used throughout the paper? Did the authors encounter toxicity in the MV4-11 cells?

There are several key reasons why we chose to use HEK293T cells for the experiment depicted in **Figure 3A**. Firstly, 293T cells can be cultured in large quantities, which is essential for our biochemical purification. Additionally, these cells are highly amenable to transfection, allowing us to achieve significant levels of bait expression. This is crucial for our objective to purify the native DCAF15 complex, while attempts to express DCAF15 at high levels in MV4-11 cells via retro- or lenti-viral delivery did not yield meaningful results.

7) Is there a way to establish whether there is a competition between DCAF15 and RAD21 that could destabilize the cohesin ring?

Our findings indicate that although RAD21 binds to the SMC1A head domain in a manner similar to DCAF15, it is essential for the interaction between DCAF15 and SMC1A. Specifically, the depletion of RAD21 results in the dissociation of SMC1A from DCAF15, as illustrated in **Figure S4C**. This observation was further supported by new data where overexpressing RAD21 enhanced the interaction between DCAF15 and SMC1A, reinforcing a model where RAD21 (and the cohesin ring assembly) is necessary for DCAF15 recruitment (**Figure S4D**). Additionally, it is known that RAD21 can bind to SMC1A through various interfaces, suggesting that DCAF15 might trigger a conformational alteration at the RAD21-SMC1A interface^{1,2}. We are considering this hypothesis, and further structural studies are planned to elucidate this interaction mechanism in a subsequent manuscript.

8) DCAF15 KO cells are definitively sensitive to DNA damaging agents. The higher sensitivity to the PARP inhibitor olaparib indicates a putative deficiency in Homologous Recombination (HR), as HR-deficient cells rely on PARP to repair DNA damage. Furthermore, DNA damage occurring at replication forks is mainly repaired via HR. Since cohesins have now been involved in DNA damage responses to favor the use of sister chromatids, and stall replication

in order to prevent amplification of damaged DNA, it would be interesting to further investigate that aspect.

We thank the Reviewer for highlighting this aspect; we concur that a defect in DNA damage sensing is evident in *DCAF15*-KO cells. Our new data show a significant increase in RAD51 abundance on chromatin in proliferating *DCAF15*-KO cells upon olaparib treatment, indicative of elevated levels of single-stranded DNA and inability to resolve stalled DNA replication forks (**Figure 2** in this letter).

Additionally, we observe increased γ H2AX, as demonstrated by immunofluorescence (IF) in **Figure S7C** and by Western blot in **Figure 7C**, suggesting that the loss of DCAF15 triggers a DNA damage response. These findings are consistent with the observed sensitivity to olaparib (Figure 7E).

Figure 2. Left, HEL cells stably expressing sgRNA targeting ROSA26 or DCAF15 were stained for the indicated proteins. Where indicated, cells were treated with olaparib (5uM) for 16 hours. Right, quantification of RAD51 intensity. Data pooled from 2 independent replicates. Two-tailed unpaired t tests (with Welch's correction), mean intensities indicated.

Minor points:

1) I would suggest better coordination of color coding throughout the manuscript to ease interpretation of various figures. All figures are now color coded to ease data interpretation.

2) The use of an endogenously tagged DCAF15 would strengthen a lot of the biochemistry experiments described. We have endogenously tagged DCAF15 and conducted several experiments that are discussed in our

response to Reviewer #3 (see points #2 and #7).

3) The introduction could be significantly shortened. We have shortened the introduction from 684 words to 501 words.

Reviewer #2 (Remarks to the Author):

The Reviewer #2 points out that “These findings are novel and significant and suggest novel functions of DCAF15 in sustaining AML proliferation via post-translational control of cohesion dynamics. The experiments are logical, clearly explained and high quality. The conclusions drawn from the results are justified and quite thought provoking. I recommend publication with minor questions.”

She/He points out the following minor point:

Minor point:

The authors checked DCAF15 knockout by detecting RBM39 degradation due to no effective commercial antibody against the DCAF15. However, published journals (for example, PMID: 33833131) clearly show a DCAF15 protein levels using DCAF15 antibodies (cat no. SAB1103260, Sigma). It would be great if the author can detect DCAF15 directly using above antibodies. If the antibody is available, it would be nice if the protein level or subcellular localization of DCAF15 can be changed during cell cycle.

Figure 3. Lysates from MV4-11 cells infected with lentiviruses expressing doxycycline inducible shRNA targeting DCAF15 (Left), HA-DCAF15 (Middle) or sgRNA targeting ROSA26 (negative control) or *DCAF15* (Right), were analyzed by western blot for the indicated proteins. Where indicated, cells were treated with doxycycline (1.0µg/ml; 24h (“+”) or 48h (“++”)) and/or indisulam (3µM; 6h).

We appreciate the Reviewer for highlighting this issue. The development of an antibody that recognizes endogenous DCAF15 has indeed been a significant challenge in studying this protein. We have tested the antibody recommended by the Reviewer, as shown in **Figure 3** in this letter. Unfortunately, this antibody fails to detect endogenous DCAF15 (Left and Right panels) as well as overexpressed DCAF15 (Middle panel).

Throughout this review process, we have successfully engineered two cell lines (OCI-AML3 and HEK293 cells) to endogenously express a FLAG-HA-FKBP12(F36V)-tagged version of DCAF15 (**Figure S2A, 5C, and 5D**). This marks the first study to detect endogenous DCAF15 using this technique, which represents a notable advancement in the field at this time.

Reviewer #3 (Remarks to the Author):

The Reviewer #3 points out that “The overall manuscript presents some interesting results although the main mechanistic insight needs further solid evidence”. She/He points out the following concerns:

Major concerns:

1) Fig. 3G and H, AlphaFold and IP results show that DCAF15 interacts SMC1A C-terminus. Is this interaction direct? For instance, confirm it by pulldown assays. We have performed a pull down assay with *in vitro* translated HA-DCAF15 incubated with FLAG-SMC1A immobilized on FLAG beads (**Figure S3C**). The data confirms that DCAF15 binds directly to SMC1A.

2) Fig. S3D and E, Both DCAF15 and RAD21 use similar alpha-helix to bind at the same site on SMC1A C-terminal domain. If so, does DCAF15 compete with RAD21 for SMC1A binding and interfere with cohesin complex formation? The above two points raise another concern together with IP results shown in Fig. 4A, Fig. S4C, depletion of SMC3 or RAD21 decreased interaction of SMC1A with DCAF15, i.e., DCAF15 accesses cohesin proteins assembled on the intact ring-shape structure. At least, the authors should explain how DCAF15 accesses SMC1A head domain which is directly bound by RAD21 in the cohesin complex. Our findings indicate that although RAD21 binds to the SMC1A head domain in a manner similar to DCAF15, RAD21 is essential for the interaction between DCAF15 and SMC1A, as noted by the Reviewer. Specifically, the depletion of RAD21 results in the disassociation of SMC1A from DCAF15, as illustrated in **Figure S4C**. This observation was further supported by new experiments where overexpressing RAD21 enhanced the interaction between DCAF15 and SMC1A (**Figure S4D**), reinforcing a model where RAD21 (and the cohesin ring assembly) is necessary for DCAF15 recruitment. Additionally, it is known that RAD21 binds to SMC1A through multiple SMC1A amino acid residues (including E1192, Y1194, D1219, and D1225)¹, as well as other cohesin proteins including SMC3 and STAG1/2, through various interfaces², suggesting that DCAF15 might trigger a conformational alteration at the RAD21-SMC1A interface. We are considering this hypothesis, and further structural studies are planned to elucidate this interaction mechanism in a subsequent manuscript. We have a comment in the Discussion section of the text about this model of interaction: “Therefore, while one could speculate that DCAF15 physically competes with RAD21 for binding to SMC1A, RAD21 makes additional contacts with the rest of the cohesin complex, including interfaces with SMC3 and STAG1/2. Thus, the α -helix of DCAF15 does not necessarily dislodge RAD21 from the cohesin complex entirely; instead, the E3 may simply remodel the intact ring-shaped assembly.”

Related to this, in Fig. 5A, SMC3ac blocks DCAF15 binding to SMC1A, HDAC8-mediated SMC3-deacetylation, occurring in G2/M phases, precedes DCAF15 binding to the cohesin complex. HDAC8 de-acetylates cohesin which then released from chromosome in anaphase. RAD21 is cleaved by separase in anaphase. The author proposed that DCAF15 interact with SMC1 relies on the full cohesin complex. All these pieces should be put together. These points are crucial to our understanding of DCAF15 binding to cohesin. Firstly, we've determined that acetylated cohesin does not bind to DCAF15 (**Figure 5A**). This suggests that DCAF15 does not target DNA-bound cohesin, as SMC3-acetylation predominantly occurs on DNA^{3,4}. We created an endogenous FLAG-HA-FKBP12(F36V)-tagged DCAF15 cell line and examined DCAF15's binding to SMC1A throughout the cell cycle and demonstrated that

SMC1A binds to endogenous DCAF15 throughout G1/S, G2, and prometaphase (**Figure 5C**), with DCAF15 levels showing a tendency to rise in G2.

Furthermore, we investigated the levels of PDS5A and CDC5A following DCAF15 degradation (upon dTAG treatment of only 6 hours) during G1/S, G2, and prometaphase. While there was an upregulation of PDS5A and CDC5A throughout the cell cycle following dTAG-induced DCAF15 degradation, the greatest increases of these proteins occurred during G2 and prometaphase (**Figure 5C**). These findings suggest that as cells progress to G2/M, and before anaphase, most of cohesin-bound PDS5A and CDC5A are released from DNA and targeted for degradation by DCAF15. This process is likely crucial for clearing the cohesin ring, preparing it for the next G1 phase⁵. Based on the data of RAD21 being essential for DCAF15-SMC1A interaction and the role of DCAF15 in promoting degradation of cohesin-bound PDS5A and CDCA5 in G2 and prometaphase, we predict that RAD21 cleavage at anaphase by Separase represents a DCAF15-independent regulation and likely fits with APC/Cyclosome induced degradation of CDCA5⁶.

3) DCAF15-cohesin interaction mutants should be phenotyped to validate the cellular defects observed in DCAF15 KD are due to interaction with cohesion.

We have investigated the possibility of generating a mutant of SMC1A that would be impaired in binding to DCAF15 but not RAD21. Our deletion approaches identified a critical region for DCAF15 interaction between amino acids 1174 and 1214. However, these truncations also disrupted interaction with RAD21. Based on the AlphaFold structure, we performed single amino acid mutagenesis which pinpointed Y1204 as a key binding site, affecting DCAF15 interaction with SMC1A but not RAD21 (**Figure S3G**). However, all these mutants only partially impaired interaction of DCAF15 with SMC1A. Currently, we lack an SMC1A mutant that exclusively disrupts interaction with DCAF15 while preserving its interaction with RAD21. Further structural studies are essential to elucidate the intricate

interaction dynamics between DCAF15 and SMC1A.

Figure 4. Lysates from MV4-11 (Left) or OCI-AML3 (Right) cells infected with lentiviruses expressing doxycycline-inducible HA-DCAF15 were analyzed by Western blot for the indicated proteins. Where indicated, cells were treated with doxycycline (1.0µg/ml; 24h (“+”) or 48h (“++”).

4) Fig. S4A and B, as well as Fig. S3A, the authors exclude SMC1 and p53 as the substrates of DCAF15 E3 by DCAF15 KD. Controls? An alternative way is to test the effects by overexpression DCAF15. Different substrates may require different threshold levels of E3. Because these are critical mechanistic points, we should be very cautious to exclude them by a single experiment.

We have conducted the experiment requested by the Reviewer. For this, we generated doxycycline-inducible DCAF15-expressing OCI-AML3 and MV4-11 cell lines (**Figure 4** in this letter). Upon doxycycline treatment, which led to the

upregulation of DCAF15, there were no changes in the levels of SMC1A or p53. This result is consistent with the data provided from our loss of function studies (**Figure S3A** and **S4A**).

5) Fig4A and B, are total levels of PDS5A and CDCA5 changed upon DCAF15 KD? We did not observe changes in the overall levels of PDS5A in total cell lysates following DCAF15

depletion, as shown in **Figure 4B** (Input); instead, levels of CDCA5 were upregulated in the total cell lysate upon DCAF15 deletion (**Figure 4B**, Input). Further, we detected a striking upregulation of both SMC1A-bound PDS5A and CDCA5 in *DCAF15*-KO cells, as illustrated in **Figure 4B**, SMC1A-IP. A similar upregulation was observed following dTAG treatment, as shown in **Figure 5C** (see also point#2). These data are in line with our interpretation that DCAF15 regulates the levels of SMC1A-bound PDS5A and CDCA5. *PDS5 protein level during cell cycle should be checked, it has been shown stable during cell cycle (Naif Al-Jomah, 2020)*. Total PDS5A levels remain constant throughout the cell cycle, as shown in **Figure 5C** and **5D**, which is in agreement with the literature referenced by the Reviewer. However, the amount of PDS5A bound to SMC1A decreases during prometaphase. This reduction is reversed following the loss of DCAF15 (**Figure 5C**), suggesting that DCAF15 plays a role in the regulation of SMC1A-bound PDS5A during this specific phase of the cell cycle.

6) In **Figure 4C** and **E**, only PDS5A, but not CDCA5, is detected with poly-Ubi, indicating that CDCA5 instability may be an indirect downstream effect of PDS5A degradation. CDCA5 degradation is known to be mediated by multiple sites phosphorylation then by the APC complex. Moreover, did the authors test WAPL, which is also mediated by PDS5A?

The Reviewer is correct that we detected ubiquitination of PDS5A but not CDCA5. Hence, the stabilization of cohesin-bound CDCA5 in *DCAF15*-KO cells may be indirectly achieved through its interaction with PDS5A, potentially protecting it from APC-Cdh1 mediated degradation⁶. We acknowledge this in our Results section with the following comment: "*This finding also suggests that the increased stability of cohesin-bound CDCA5 in DCAF15-knockout cells could be a downstream result of loss of DCAF15-dependent PDS5A ubiquitination and removal from the cohesin complex, supporting a model by which sustained interaction of CDCA5 with cohesin-bound PDS5A may indirectly shield it from APC-Cdh1-mediated degradation.*"

Furthermore, we investigated the ubiquitination of WAPL and found no evidence of ubiquitination (**Figure 4E**). This result aligns with the observed stable levels of both total and cohesin-bound WAPL in *DCAF15*-KO cells (**Figure 4B**).

7) **Fig. 5B**, the authors conclude that DCAF15 binds to cohesin that has been de-acetylated by HDAC8 and catalyzes ubiquitylation/degradation of PDS5A and subsequent release of CDCA5; thus, refreshing cohesin for ESCO1/2-mediated re-acetylation upon reloading onto chromatin. This conclusion is important and needs further evidence, for instance, using an inducible degron system like AID2, to deplete DCAF15 in G2/M and examine the SMC3 and SMC3ac levels in the next cycle. We appreciate the opportunity to clarify our findings regarding the impact of DCAF15-depletion. We created cells with endogenously-tagged FLAG-HA-FKBP12(F36V)-DCAF15. We induced cell synchronization in G2 phase using Ro-3306 and then triggered the degradation of endogenous DCAF15 by treating the cells with dTAG-13 for 6 hours. Notably, the degradation of DCAF15 during G2 phase did not influence SMC3-acetylation, as illustrated in **Figure 5C**.

However, when we assessed SMC3-acetylation in the next cell cycle phase by releasing cells from the Ro-3306 block, we observed a distinct acetylation defect in SMC3 within the DCAF15-depleted cells (**Figure 5D** and **S5C**). This observation suggests that the acetylation defect may stem from the improper degradation of PDS5A and CDCA5 during the G2/M transition in cells lacking DCAF15, hampering re-acetylation in the following cell cycle. *This system is also important for experiments shown in Fig7. PDS5A and CDCA5 are recruited to*

cohesin in a SMC3ac-dependent way, does deacetylation also contribute to their dissociation and DCAF enhances this process? SMC3 acetylation is known to facilitate the recruitment of PDS5A and CDCA5 to the cohesin complex⁴. Our findings introduce an additional layer of complexity, indicating that in synchronized cells, as they progress into prometaphase, SMC3-acetylation is reduced and so are the levels of SMC1A-bound PDS5A and CDCA5, in a DCAF15-dependent manner. This observation is significant as it suggests that the detachment of these proteins from the cohesin complex, traditionally observed during mitosis, is attributable to their degradation.

8) Fig. 5D-G, did the authors examine ESCO2/1 simultaneously? This will tell us how DCAF15 affects SMC3ac, e.g., DCAF15 blocks their recruitment? We have assessed the recruitment of ESCO1 in *DCAF15*-KO cells by ChIP-qPCR (**Figure S5G**). This new data displays a defect in ESCO1 recruitment upon DCAF15 loss, in line with the defect in acetylation of SMC3 shown in *DCAF15*-KO cells.

Figure 5. Cas9+ MV4-11 *DCAF15*-WT (sgROSA26) and *DCAF15*-knockout (sgDCAF15#2) cells were cultured in the presence of caffeine at the indicated concentrations. Relative sgRNA+ proportions were assessed by flow cytometric analysis and normalized to day 2 post-sgRNA-infection.

9) Fig7C, Is ATM-CHK2 activated? Can ATM and CHK2 inhibitors rescue *DCAF15* KO cells like p53 inactivation? We reloaded samples from MV4-11 cells with sgROSA26 and sgDCAF15 and assessed the levels of phosphorylated CHK2 (pCHK2), ATM (pATM), as well as Cleaved-Caspase-3 and p21 (asked by Reviewer#4), as

depicted in **Figure 7** in this letter. Data quantification reveal that both pCHK2 and pATM levels are elevated in *DCAF15*-KO cells, although the change in pATM only reached a p-value of 0.112.

We attempted a rescue experiment using caffeine (an ATM/ATR inhibitor). The experiment is shown in this letter (**Figure 5** in this letter) and reveals that ATM/ATRi did not lead to a successful rescue of cell proliferation upon *DCAF15* loss. Hence, we interpret that ATM/ATR may not be the only kinases involved in the activation of the p53 pathway in *DCAF15*-knockout cells.

10) Does *DCAF15* functions in sister chromatid cohesion or separation? We conducted metaphase spread analysis and observed that chromosomes in *DCAF15*-knockout cells display increased propensity to form railroad chromosomes, indicating a potential defect in sister chromatid cohesion, as shown in **Figure 6** in this letter. This finding suggests that *DCAF15* plays a role in maintaining proper chromosome structure and function during cell division.

Minor issues:

11) Figure S1A does not clearly show which dot is *DCAF15*. We changed the color of the dot to blue and added a line to label *DCAF15*.

Figure 6. Cas9+ HEL cells were infected with sgROSA26 or sgDCAF15 and treated with colcemid at 0.1ug/ml for 3 hours. Left, representative example of metaphase spread. Right, quantification of railroad chromosomes (indicated with yellow arrows). Data pooled from 3 independent replicates. Two-tailed unpaired t test (with Welch's correction),

Reviewer #4 (Remarks to the Author):

The Reviewer #4 points out that “The authors of the study present here a novel role for DCAF15 in regulating the cohesin complex”. She/He points out the following concerns:

Please see comments below:

1) The number of differentially regulated genes captured by the RNA-seq seems very low fig2A and 2E, could the authors comment? This comment has also been issued by Reviewer #1 (please also see Response to Reviewer #1, point #2).

The RNA-sequencing experiment shown in Figure 2A was intentionally conducted 4 days post-infection to ensure to capture the earliest transcriptional changes upon DCAF15 loss and before it would interfere with cell proliferation, as detailed in Figures 1F, 1H and 2H. Upon Reviewer #1's suggestion, we modified OCI-AML3 cells to express an N-Terminal FLAG-HA-FKBP12(F36V) fusion protein at the endogenous DCAF15 locus to allow dTAG--mediated DCAF15 degradation (Figure S2A). Subsequent RNA-sequencing at 24 and 72 hours after dTAG treatment revealed a significant early upregulation of the *TP53* pathway, becoming more pronounced at 72 hours (Figure S2C). These observations are consistent with our previous CRISPR-mediated *DCAF15*-knockout studies (Figure 2A), indicating that p53 activation is an immediate consequence of DCAF15 depletion.

2) Fig2D WB, it is not clear how many times this experiment was performed, 3 independent experiments are expected with quantification. We reloaded samples from three independent experiments of MV4-11 cells with sgROSA26 and sgDCAF15 and assessed the levels of Cleaved-Caspase-3 and p21. Data quantification revealed that both Cleaved-Caspase-3 and p21 levels are significantly elevated in *DCAF15*-KO cells (Figure 7 in this letter).

Figure 7. Left, lysates from 3 independent experiments of MV4-11 cells infected with lentiviruses expressing sgRNA targeting ROSA26 (negative control) or *DCAF15* were analyzed by Western blot for the indicated proteins. Right, for the indicated proteins, ImageJ quantifications (normalized to sgROSA26 and to Vinculin protein loading control quantification) were plotted (n=3, t-test, two tailed, unpaired).

3) Fig2G apoptosis is increased by less than 2% in *DCAF15* KO, it is unlikely to explain the two fold differences observed in the proliferation assay. Cell cycle analyses would reveal whether the phenotype is mostly due to an impact on cell cycle regulation. This point was also raised by Reviewer#1 (please see also the response to Reviewer#1, point #5).

The proliferation assays shown in Figure 2F (as well as Figure 1F) illustrate a proliferative competition assay in which infected cells (mCherry-positive) are mixed with

non-infected cells (mCherry-negative) and respective ratios are monitored over time. Cells are split at a 1:10 ratio every 3 days, which naturally favors cells with higher proliferation potential (mCherry-negative). Consequently, after several passages, the ratio of mCherry-

positive to mCherry-negative cells decreases. Over a span of 25 days, this assay demonstrates that non-infected cells can outcompete the infected ones.

Since this assay measures the cell-cell proliferation competition in a mixed culture, it doesn't provide a direct representation of cell proliferation. To more directly examine the proliferation deficit caused by DCAF15 loss, we sorted MV4-11 cells infected with either sgROSA26 or sgDCAF15 and conducted a non-competitive proliferation assay (**Figure S1H**). In this assay, cells are counted and replated at the original density every 3 days, offering a more direct measure of proliferation impairment. Through this additional method, we confirmed that the loss of DCAF15 results in a significant reduction in cell proliferation over time. We acknowledge the Reviewer's insight that apoptosis alone may not fully account for the observed proliferation defect in *DCAF15*-KO cells. We have repeated the cell cycle analysis in MV4-11 cells infected with sgROSA26 and sgDCAF15 and found no major cell cycle impairment (**Figure S7A**).

Based on these observations, we propose that the proliferation defect upon DCAF15 loss is likely attributable to a combined effect of decreased cell proliferation together with apoptosis induction. This revised understanding is now incorporated into the text as following: *"Altogether, these results demonstrated that DCAF15 loss suppresses AML via activation of p53 tumor suppressive programs, and that the growth defect of AML cells lacking DCAF15 may be attributable to a combination of both reduced cell proliferation rate as well as induction of apoptosis."*

Figure 8. HEK293T stably expressing HA-DCAF15 were treated with DMSO (-LMB) or leptomycin B (+LMB) for 4 hours. Staining for DAPI (Left), HA (Middle), and Merge of both (Right) are shown.

4) DCAF15 is expected to be mainly expressed in the cytoplasm and not in the nucleus where the cohesin complex is found. In order to confirm that DCAF15 plays a role on the cohesin complex, nuclear localization of DCAF15 should be demonstrated by immunofluorescence experiments. These experiments would also allow the demonstration of the co-localization in the nucleus of DCAF15 with different cohesin proteins. We evaluated the localization of virally transduced HA-DCAF15 in cells (**Figure 8** in this letter). DCAF15 shows a primarily cytoplasmic staining, however a short treatment

with leptomycin B (LMB) promotes nuclear accumulation of DCAF15. This data suggests that DCAF15 constitutively shuttles from the cytoplasm into the nucleus. We have tried co-localization with cohesin members SMC1A and SMC3, however diffuse nuclear positive staining of these abundant proteins renders it difficult to derive definitive co-localization measurement. Nevertheless, the nuclear localization of DCAF15 is in line with its role in binding SMC1A.

It would be interesting to observe whether protein or RNA levels of the different components of the cohesin complex are affected by DCAF15 KO? Cohesin complex component mRNA

levels are shown **Figure S4B**. The data show no significant changes in the mRNA levels of cohesin core members as well as associated proteins upon DCAF15 loss. Protein levels in total cell lysates and the SMC1A-bound pool are shown in **Figure 4B**.

5) The link between ac-SMC3 and DCAF15-destabilises cohesion-bound PDS5A and CDCA5 should be further investigated to confirm that DCAF15 impacts ac-SMC3 via its role on PDS5A and CDCA5. We present additional data indicating that the degradation of DCAF15 upon treatment with dTAG for 6 hours results in the stabilization of SMC1A-bound CDCA5 and PDS5A, particularly during G2 phase and prometaphase (**Figure 5C**). Notably, while the G2-phase degradation of DCAF15 does not alter SMC3-acetylation levels within the same cell cycle phase, a notable decrease in SMC3-acetylation is observed as cells transition into the following G1 phase (**Figure 5D** and **S5C**). This suggests that PDS5A and CDCA5 degradation during mitosis is essential for efficient SMC3-acetylation in the following cell cycle. Furthermore, our findings demonstrate a reduction in ESCO1 levels at genomic regions where SMC3-acetylation is defective in *DCAF15*-KO cells (**Figure S5G**, also see Reviewe#3, point #8). This observation implies that ESCO1 recruitment is compromised in the absence of DCAF15, indicating a crucial role of DCAF15 in facilitating ESCO1's function in the acetylation process.

Assessing the impact of DCAF15 on SMC3-acetylation via double genetic knockout (*i.e.*, sgDCAF15 + sgCDCA5 or sgDCAF15 + sgPDS5A) is not compatible with cell viability. Indeed, cohesin members SMC1A, SMC3, RAD21, CDCA5, and PDS5A are essential in AML (data from Broad Institute DepMap).

In summary, our research offers new evidence that DCAF15-mediated destabilization of SMC1A-bound CDCA5 and PDS5A during mitosis is crucial for clearing cohesin from these accessory proteins, thereby facilitating the subsequent loading of CDCA5/PDS5A-free cohesin back onto chromatin upon cell division.

6) Fig7C \$\gamma\$ H2AX should be quantified by ImmunoFluorescence (scoring the Foci). We have quantified γ H2AX foci in MV4-11 and OCI-AML3 cells upon DCAF15 loss, as shown in **Figure S7C**. In both experiments, the loss of DCAF15 resulted in a significant upregulation of γ H2AX foci.

7) Fig7D DCAF15 KO cells already have a proliferative disadvantage (see fig1F), therefore it is not clear what is the effect of CPT. A synergistic experiment should provide more insights in the cumulative or synergistic effect of CPT and DCAF15 loss. The same question is raised by the following figure using PARP inhibitors. We acknowledge and appreciate the Reviewer's insight regarding the experiment presented in **Figure 7D**. We agree that the data shown do not directly address a synergistic effect of camptothecin (CPT) or olaparib with DCAF15 loss. The primary objective of the experiments in **Figures 7D** and **7E** is to demonstrate that DNA damage-inducing drugs increase the sensitivity of AML cells to DCAF15 loss. Indeed, the data presented shows that, normalized to *DCAF15*-KO cells cultured in the presence of DMSO, *DCAF15*-KO cells cultured in the presence of camptothecin or olaparib are outcompeted by non-infected *DCAF15*-WT parental cells at significantly faster rates. This is not the case for control cells infected with sgROSA26.

In response to the reviewer's comment, we have made sure not to use the term "synergy" to reflect the nature of the interaction between DCAF15 loss and the effects of these drugs.

8) Most of the work has been carried out on one cell line the MV4-11, although the authors present strong evidences of a potential role of DCAF15 on the cohesion complex, the use of one cell line is not sufficient to confirm a pan effect of DCAF15 on AML which is a highly heterogeneous disease. In the original submission, we have assessed proliferation competition in a panel of AML cell lines (**Figure 2H**).

We have conducted further experiments to elucidate the role of DCAF15 in the non-competitive proliferation of an additional AML cell line (OCI-AML3; not carrying MLL-rearrangement or FLT3 mutations), the results of which are presented in **Figure S1H**.

Additionally, we have included data on apoptosis in additional AML cell lines (OCI-AML3 and MOLM-13), depicted in **Figure 2I**. Moreover, our findings demonstrate an increase in γ H2AX in MV4-11 and OCI-AML3 cell lines, further substantiating the impact of DCAF15 loss (**Figure S7C**). Finally, we also present transcriptional changes observed through RNA-sequencing of OCI-AML3 cells upon DCAF15 depletion, as shown in **Figure S2A-C**.

9) In addition, to determine whether DCAF15 is a safe target, similar experiments should be carried out on normal hematopoietic stem/progenitor cells to determine tolerance and toxicity. This is a valuable observation that we are eager to investigate further. We are in the process of creating mouse models to specifically ablate *Dcaf15* in bone marrow. Additionally, we are developing PROTAC molecules aimed at inducing degradation of the DCAF15 protein, which could have therapeutic implications. Given the extended duration required for these experiments, we consider this aspect beyond the current study's scope. We intend to address this in a subsequent manuscript, where we will present these findings in detail.

We thank all four Reviewers for their time and dedication in reading and providing valuable feedback to improve our manuscript.

Sincerely,

Luca Busino, PhD
Associate Professor, Department of Cancer Biology
Associate Investigator, Abramson Family Cancer Research Institute
Member, Abramson Cancer Center Tumor Biology Program
Core Member, Penn Center for Genomic Integrity
Perelman School of Medicine
University of Pennsylvania
[Email: businol@upenn.edu](mailto:businol@upenn.edu)
Office: 705 BRBII/III
Phone: 215-746-2569

References

1. Deardorff, M. A. *et al.* RAD21 mutations cause a human cohesinopathy. *Am J Hum Genet* **90**, 1014–1027 (2012).
2. Shi, Z., Gao, H., Bai, X.-C. & Yu, H. Cryo-EM structure of the human cohesin-NIPBL-DNA complex. *Science (1979)* **368**, 1454–1459 (2020).
3. Beckouët, F. *et al.* An Smc3 acetylation cycle is essential for establishment of sister chromatid cohesion. *Mol Cell* **39**, 689–699 (2010).
4. Ruiten, M. S. *et al.* The cohesin acetylation cycle controls chromatin loop length through a PDS5A brake mechanism. *Nat Struct Mol Biol* **29**, 586–591 (2022).
5. Deardorff, M. A. *et al.* HDAC8 mutations in Cornelia de Lange syndrome affect the cohesin acetylation cycle. *Nature* **489**, (2012).
6. Rankin, S., Ayad, N. G. & Kirschner, M. W. Sororin, a Substrate of the Anaphase-Promoting Complex, Is Required for Sister Chromatid Cohesion in Vertebrates. *Mol Cell* **18**, 185–200 (2005).

REVIEWERS' COMMENTS

Reviewer #1 (Remarks to the Author):

The authors have addresses all of my concerns. I congratulate them on such an interesting and impactful piece of work.

Reviewer #2 (Remarks to the Author):

This revised manuscript is clearly explained followed by all reviewer's points. The experiments are also logical with high quality. The conclusion drawn from the results are justified and quite thought provoking. I recommend publication of revised manuscript.

Reviewer #3 (Remarks to the Author):

The authors have addressed major concerns with a bunch of nice new data, especially those from endogenous tagged DCAF15 and degron. I support the publication, with two minor suggestions:

1. reviewer 3, point 6, the answer reminds me whether wapl depletion will rescue the phenotypes of DCAF15 KD. It will significantly improve the main model of this study, right?

2. reviewer 4, point 4, maybe try PLA rather than IF?

Reviewer #4 (Remarks to the Author):

The authors have addressed most of my comments, providing new data where needed and answering point by point my questions in a sound manner.

Reviewer Comments:

Reviewer #1 (Remarks to the Author): The Reviewer #1 states “*The authors have addressed all of my concerns. I congratulate them on such an interesting and impactful piece of work.*”

We thank Reviewer #1 for the valuable feedback that helped to improve the manuscript.

Reviewer #2 (Remarks to the Author): The Reviewer #2 states “*This revised manuscript is clearly explained followed by all reviewer’s points. The experiments are also logical with high quality. The conclusion drawn from the results are justified and quite thought provoking. I recommend publication of revised manuscript.*”

We thank Reviewer #2 for the insightful comments.

Reviewer #3 (Remarks to the Author): The Reviewer #3 states “*The authors have addressed major concerns with a bunch of nice new data, especially those from endogenous tagged DCAF15 and degran. I support the publication, with two minor suggestions:*

- 1. reviewer 3, point 6, the answer reminds me whether wapl depletion will rescue the phenotypes of DCAF15 KD. It will significantly improve the main model of this study, right?*
- 2. reviewer 4, point 4, maybe try PLA rather than IF?”*

We thank Reviewer #3 for the insightful review which helped to strengthen the manuscript. With regard to Point #1 raised by Reviewer #3 (*Reviewer 3, point 6, the answer reminds me whether WAPL depletion will rescue the phenotypes of DCAF15 KD. It will significantly improve the main model of this study, right?*), in our study, we show that DCAF15 does not promote WAPL degradation nor ubiquitylation (**Figure 4B** and **4E**). These data align with the observed stable levels of both total and cohesin-bound WAPL in *DCAF15*-KO cells (**Figure 4B**). Because no change in WAPL is detected upon *DCAF15* loss, *WAPL* co-knockout with *DCAF15*-KO is not the ideal experimental condition to rescue the cell toxicity phenotype induced by *DCAF15* loss. Perhaps the Reviewer meant *PDS5A* rather than *WAPL*. If that is the case, we have tried to perform a double-knockout of *PDS5A* and *DCAF15*. Unfortunately, as with many components of the cohesin complex, deletion of *PDS5A* is lethal to cells making this experiment not feasible.

Point #2 raised by Reviewer #3 (*reviewer 4, point 4, maybe try PLA rather than IF?*) is a valuable suggestion and we are planning to pursue it in the future. In fact, we would like to use PLA experiments to analyze the interaction of *DCAF15* and *SMC1A* throughout the cell cycle to pinpoint the exact phase where this interaction occurs. We hope the Reviewer agrees that these experiments will require substantial effort to determine the specificity of the PLA staining. Our intention is to include this research area in a follow-up manuscript without impacting the publication timeline of the current study.

Reviewer #4 (Remarks to the Author): The Reviewer #4 states “*The authors have addressed most of my comments, providing new data where needed and answering point by point my questions in a sound manner.*”

We thank Reviewer #4 for the valuable comments that helped to improve the manuscript.

We thank all four Reviewers for their time and dedication in reading and providing valuable feedback to improve our manuscript.